# An improved fluorescent tag and its nanobodies for membrane protein expression, stability assay, and purification

Hongmin Cai [1,3], Hebang Yao [1,3], Tingting Li [1,3], Cedric A. J. Hutter [2], Yanfang Li[1], Yannan Tang [1], Markus A. Seeger [2] & Dianfan Li [1✉]

Green fluorescent proteins (GFPs) are widely used to monitor membrane protein expression, purification, and stability. An ideal reporter should be stable itself and provide high sensitivity and yield. Here, we demonstrate that a coral (*Galaxea fascicularis*) thermostable GFP (TGP) is by such reasons an improved tag compared to the conventional jellyfish GFPs. TGP faithfully reports membrane protein stability at temperatures near 90 °C (20-min heating). By contrast, the limit for the two popular GFPs is 64 °C and 74 °C. Replacing GFPs with TGP increases yield for all four test membrane proteins in four expression systems. To establish TGP as an affinity tag for membrane protein purification, several high-affinity synthetic nanobodies (sybodies), including a non-competing pair, are generated, and the crystal structure of one complex is solved. Given these advantages, we anticipate that TGP becomes a widely used tool for membrane protein structural studies.

[1] University of Chinese Academy of Sciences, National Center for Protein Science Shanghai, CAS Center for Excellence in Molecular Cell Science, Shanghai Institute of Biochemistry and Cell Biology, Chinese Academy of Sciences, 320 Yueyang Road, 200031 Shanghai, China. [2] Institute of Medical Microbiology, University of Zurich, Zurich, Switzerland. [3] These authors contributed equally: Hongmin Cai, Hebang Yao, Tingting Li. ✉email: dianfan.li@sibcb.ac.cn

Current strategies for membrane protein structural studies require functional isolation in large quantities but their hydrophobicity brings great challenges in almost every step, such as expression, purification, and crystallization or cryo-electron microscopy[1–8]. Overcoming these challenges often relies on multilevel optimization. To increase expression or reduce degradation, homologs and mutants sometimes in dozens to hundreds[9–13], as well as hosts[14–22] and culture conditions[23–27], are evaluated. To increase stability, numerous constructs by scanning mutagenesis or rational design are screened[28–35]. To find supportive purification conditions, detergent type[36–41], lipid additive type and ratio[42–44], as well as salts[45], ligands[46], and pH[47,48], are assessed[9,37,49]. Such optimization brings a large number of screen variables, which can be time-consuming and costly. Increasing throughput is crucial.

Fluorescent proteins, mostly green fluorescent proteins (GFPs), fit this purpose as a fusion partner for the study of protein of interest (POI). Its bright fluorescence in the visible wavelengths allows real-time monitor of expression level, folding, chromatography efficiency and profile, and stability of membrane proteins[27,46,50]. Of particular interest is the fluorescence-detection size exclusion chromatography (FSEC)-based thermostability assay (FSEC-TS)[46] for measuring apparent melting temperature ($T_m$). This method is generally applicable; i.e., does not require specific enzymatic or ligand-binding assays. An additional advantage is that the assay can be done without purification, the failure of which is sometimes the very reason for stability assays in the first place. Since its introduction to the field in 2012, FSEC-TS has been used successfully, along with mutagenesis, for the thermostabilization of several membrane proteins for structural studies[46,51–54]. With an automated HPLC system, ~100 constructs/conditions can be screened in an overnight run.

One limitation of the most commonly used GFPs, however, is their modest thermostability. It fails to serve as a reporter for FSEC-TS of membrane proteins with apparent $T_m$ exceeding the $T_m$ of GFP itself which is ~76 °C (10-min heating)[46]. While this appears to be reasonably high, it is important to note that the actual $T_m$ may be far below the apparent $T_m$ (or more appropriately termed as $T_{aggregation}$) because proteins can undergo denaturation long before aggregation[37,55]. Therefore, sometimes conditions leading to apparent $T_m$ of higher-than-76 °C need to be sought. FSEC-TS assays in such cases have relied on intrinsic tryptophan fluorescence[46]. This inevitably requires the purification of POIs, which could be challenging for membrane proteins. A more stable fluorescent protein for FSEC-TS assays at high temperatures will be welcome.

We were curious to see whether a thermostable GFP (TGP, from coral)[56–58] can fulfill this demand. Here, our systematic survey study with several membrane proteins demonstrates that TGP is indeed a robust and reliable FSEC-TS reporter as it enables assays at ~90 °C with $T_m$ values in close agreement with established methods. Our data also show improved expression for TGP fusion of four test membrane proteins in four systems. High-affinity synthetic nanobodies (sybodies) against TGP have been generated using ribosome and phage display[59], enabling purification of a membrane protein to comparable purity (~90%) with that from immobilized metal affinity chromatography. The binding mode of one of the sybodies to TGP has been revealed in detail by X-ray crystallography. The overall superior characteristics of TGP over GFPs, together with the sybody tools, should encourage the use of TGP as a fusion partner for not only membrane proteins but may also for soluble proteins.

## Results

**TGP was brighter than the conventional GFPs.** Two GFP variants, commonly used as fusion tags in *Escherichia coli*[50,60]

(ecGFP, NCBI AGT98535.1) and *Saccharomyces cerevisiae*[27], insect cells[61,62], and mammal cells[63] (scGFP, NCBI ABI82039.1), were used for comparison. The engineered Azami-Green[56–58], although also a GFP, is referred to as TGP. In a plate reader with excitation/emission wavelength pair of 488/512 nm, the brightness ratio was 1: 0.84: 0.43 (TGP: ecGFP: scGFP) (Fig. 1a), agreeing with the fact that TGP has a higher extinction coefficient ($\epsilon_{493, TGP} = 64,000$ M$^{-1}$ cm$^{-1}$, $\epsilon_{490, ecGFP} = 49,550$ M$^{-1}$ cm$^{-1}$) and a higher quantum yield ($\Phi_{TGP} = 0.66$, $\Phi_{ecGFP} = 0.60$) than GFPs[56]. In in-gel fluorescence, TGP was as bright as ecGFP, and twice as bright as scGFP, with a ratio of 1: 0.99: 0.46 (TGP: ecGFP: scGFP) (Fig. 1b, c, Supplementary Fig. 1a). The slight discrepancies between the two methods may be due to different filter settings between the two systems, or possible differences in their response to the chaotropic sodium dodecyl sulfate (SDS) in the electrophoresis buffer. In addition, the scGFP contains mutations[27] that cause yellow-shift of the excitation wavelength ($\lambda_{max} = 502$ nm), which is partly responsible for its relatively low intensity in both counting and in-gel fluorescence analysis.

**TGP was more stable than GFPs.** As expected, TGP displayed the highest $T_m$ at 95.1 °C, which was 18.0-°C above ecGFP and 26.7-°C above scGFP (Fig. 1d). Because membrane proteins are generally solubilized in detergents for purification, we also measured their $T_m$ in dodecylmaltoside, a mild detergent, and lauryldimethylamine oxide, a harsher zwitterionic detergent, at 1%(w/v) concentration. The $T_m$ for fluorescent proteins decreased by 1.8–2.6 °C in dodecylmaltoside (Fig. 1d) and by 4.9–5.3 °C in lauryldimethylamine oxide (Fig. 1d), reflecting the modest chaotropic effects of these detergents.

**TGP was a robust and reliable reporter for stability assays.** To test TGP as a reporter for stability assays, its $T_m$ was measured using FSEC-TS (Fig. 1e, Supplementary Fig. 2a) and compared to that obtained by fluorescence counting. Similar to previous findings for GFPs[43,46], the $T_m$ values by the two methods were similar (Fig. 1e), suggesting FSEC-TS was reliable for monitoring TGP unfolding. Consistent with the fluorescence counting method, TGP showed higher stability than both GFPs (Fig. 1e, Supplementary Fig. 2b, 2c). Notably, the main-peak profile remained unchanged even after heating at 99 °C (Supplementary Fig. 2d), which is an important characteristic for a reporter.

Next, we performed three sets of experiments to test its reliability as a FSEC-TS reporter. First, we compared the FSEC-TS profiles between GFP and TGP using three membrane protein fusions: a microbial (*Mycobacterium smegmatis*) homolog of the human Δ8–7 sterol isomerase (*ms*SI, 29.0 kDa, six predicted transmembrane helices) expressed in *E. coli*, and the human Δ8–7 sterol isomerase (*h*SI, 26.4 kDa, five transmembrane helices) and the human CDP-diacylglycerol-inositol 3-phosphatidyl-transferase (PIS, 23.5 kDa, four transmembrane helices) expressed in *S. cerevisiae*. As shown in Fig. 2a, b, and Supplementary Fig. 3a, 3b, the FSEC-TS melting profile for *ms*SI and PIS did not show notable changes upon the replacement of GFP by TGP. For *h*SI, the TGP signal decreased at a lower rate than scGFP, displaying a ~5-°C increase in the apparent $T_m$ (Fig. 2c, Supplementary Fig. 3c). The reason for this will be discussed.

Second, we compared the FSEC-TS profile of membrane proteins with and without TGP fusion using purified samples, which are necessary for the tryptophan-based assay. For this we tested *h*SI and its homolog from *Thermothlomyces thermophilus* (*tt*SI)[64]. As shown in Fig. 2d and Supplementary Fig. 3d, 3e, the TGP-based pseudo-melting curve for the fusion protein was nearly superimposable with the tryptophan-based curve for the

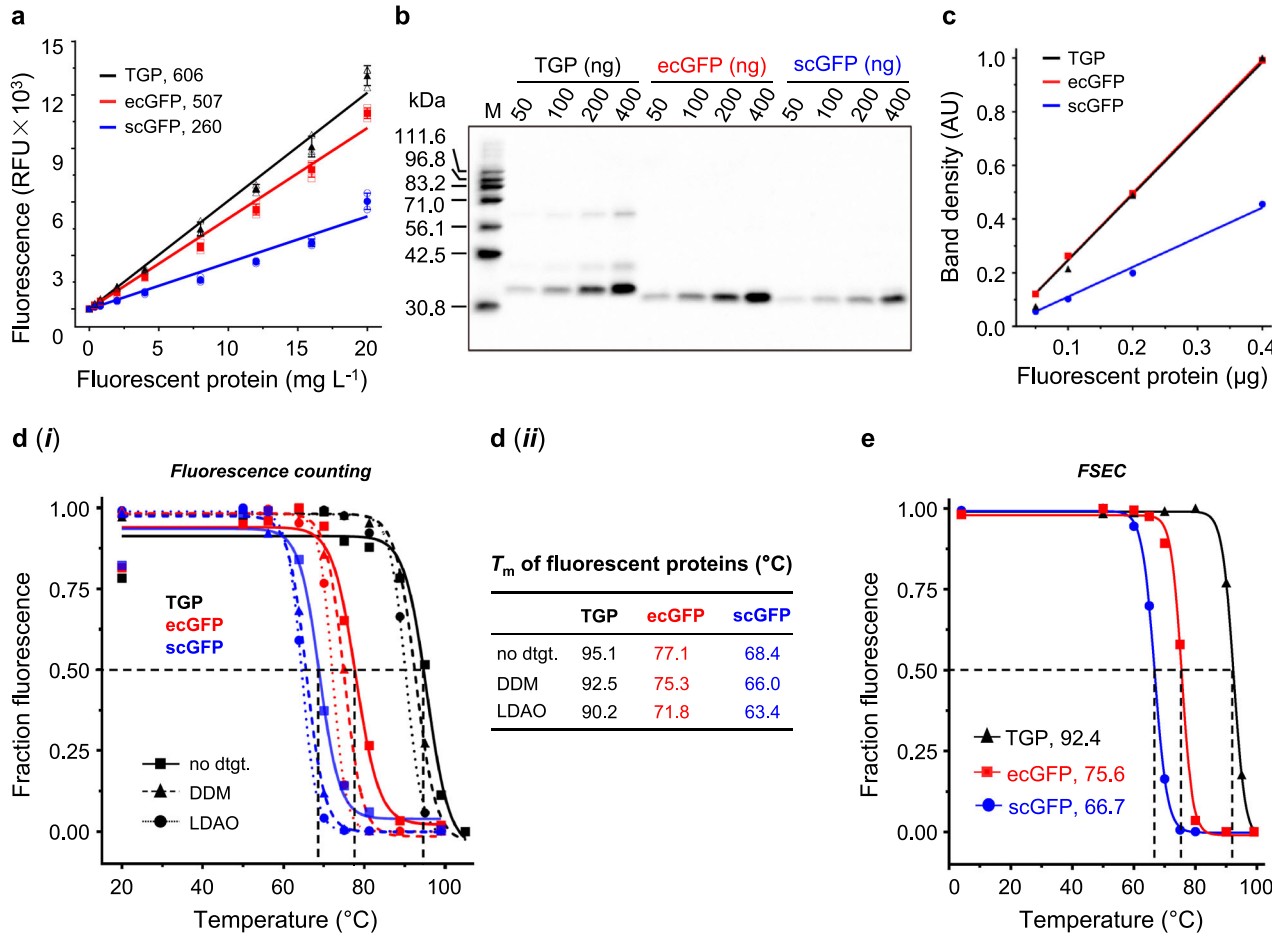

**Fig. 1 TGP was brighter and more stable than GFPs. a** Linear correlation between fluorescence (in a relative unit, RFU) and concentration (in a 0.2-mL solution) with indicated slope (RFU for 1 mg L$^{-1}$ of fluorescent protein). Mean and standard deviation of three independent experiments are plotted. **b** In-gel fluorescence image with inverted colors. M: home-made fluorescent markers[14]. Data are representative of two independent experiments. **c** Linear correlation between integrated in-gel fluorescence (in an arbitrary unit, AU) and loading using data in **b**. **d** Fluorescence counting-based melting curve (*i*) and $T_m$ value (*ii*) of the fluorescent proteins in the absence or presence of indicated detergents (dtgt). Data are from a single experiment. **e** FSEC-based melting curve of the fluorescent proteins in the presence of dodecylmaltoside. Graph depicts mean of three technical replicates. The unit (°C) for $T_m$ is omitted for all panels. In this report, all samples were heated for 20 min for $T_m$ measurement unless stated otherwise. In panels **a**, **c**, and **e**, black triangle denotes TGP, red square denote ecGFP, and blue circle denotes scGFP. In **d**, TGP traces are black, ecGFP traces are red, and scGFP traces are blue; Solid line with square denotes samples without detergents, dashed line with triangle denotes DDM samples, and dotted line with circle denote LDAO samples. DDM dodecylmaltoside, FSEC fluorescence-detection size exclusion chromatography, GFP green fluorescent protein, LDAO lauryldimethylamine oxide, TGP thermostable GFP, $T_m$ melting temperature.

TGP-free protein in both of the cases. Therefore, the TGP fusion did not change the apparent $T_m$ of *tt*SI or *h*SI.

Third, an orthogonal validation of TGP for FSEC-TS was performed using the 'gold-standard' enzymatic assay of the glycerol 3-phosphate acyltransferase from *Staphylococcus pneumonia* (*sp*PlsY, 22.9 kDa, seven transmembrane helices). The melting profiles from the two methods were overall similar but differed in fine details (Fig. 2e). Specifically, the activity dropped gradually upon heating, whilst the fluorescence increased before dropping. Compared to the enzymatic assay, the FSEC-TS curve entered the denaturation phase later but fell more quickly, resulting in a slightly lower $T_m$ (by ~5 °C). Taken together, the results demonstrated that TGP was an overall faithful FSEC-TS reporter despite slight variations.

Next, we tested the robustness of TGP for FSEC-TS assay using the ultra-stable PlsY from *Aerocus aquifex* (*aa*PlsY, 20.9 kDa, 7 transmembrane helices), which has a half-life of 30 min at 90 °C as revealed by an activity-based assay[65]. As shown in Fig. 2f and Supplementary Fig. 3f, the fluorescence of *aa*PlsY-TGP remained

unchanged at 85 °C. Because both *aa*PlsY (Fig. 2f) and TGP (Fig. 1e) started to unfold at higher temperatures, the drop of fluorescence beyond 85 °C could be a result of denaturation of either or both proteins. Thus, the apparent $T_m$ of 91.1 °C, although agreeing with the activity-based $T_m$ value, may or may not be the true apparent $T_m$ of *aa*PlsY. Nevertheless, it showed that the FSEC profile of TGP, in this case as a fusion to a membrane protein, was unchanged at 85 °C. As an additional example, we repeated the experiment with the thermostable conSI (a sterol Δ8–7 isomerase thermostabilized by consensus mutagenesis[64]). The result showed that TGP fluorescence only dropped by ~20–30% at 90 °C, reporting an apparent $T_m$ value of 94.4 °C (Supplementary Fig. 4), which was close to that of TGP itself (Fig. 1e). Again, the FSEC shape remained unchanged even at 99 °C (Supplementary Fig. 4d). Thus, fusing to membrane proteins did not change the stability or FSEC behavior of TGP. Based on this, we suggest the upper limit for TGP-based FSEC-TS assays be at 90 °C. As expected, GFPs, even the more stable ecGFP, was incapable of monitoring membrane protein unfolding

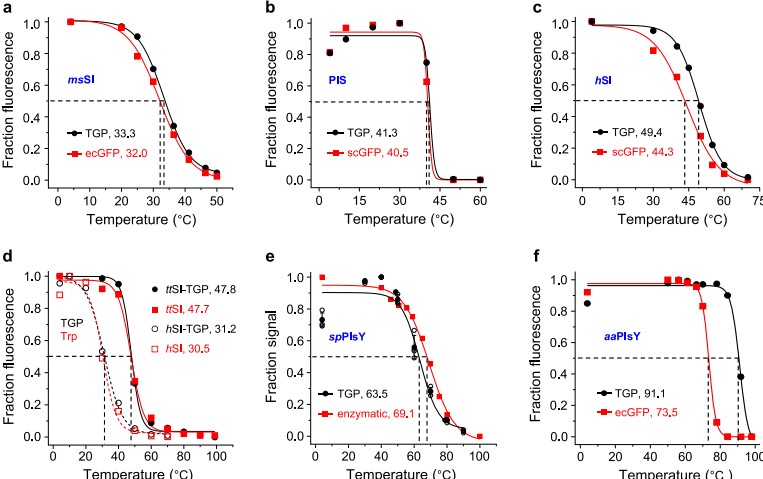

**Fig. 2 TGP was a reliable and robust reporter for FSEC-TS assay.** $T_m$ values of indicated constructs measured by different methods are shown in each panel **a**–**f**. Black circles denote data using TGP fluorescence, and red squares denote data using ecGFP (**a**, **f**), scGFP (**b**, **c**), tryptophan fluorescence (**d**), or enzymatic activity (**e**). In **d**, solid lines and close symbols denote *tt*SI samples, and dashed lines and open symbols denote *h*SI samples. Assays were performed using nonpurified samples except for **d** in which the samples had been purified using His-tag. Data are from a single experiment except for the FSEC data in **e** which are from four independent experiments. Standard deviation is not applicable for the data points (48.8 and 78.8 °C) because no replicates were performed for these two temperatures. To keep consistent with literature[65], the heating time was 30 min for *aa*PlsY. The $T_m$ difference of *h*SI-TGP between **c** and **d** is likely due to delipidation effect (see Discussion). FSEC-TS fluorescence-detection size exclusion chromatography-based thermostability assay, GFP green fluorescent protein, TGP thermostable GFP, $T_m$ melting temperature.

at such temperatures, falsely reporting an under-measured $T_m$ of 73.5 °C (Fig. 2f). Thus, the results showed TGP was an improved FSEC-TS reporter owing to its ultrastability.

**Replacing GFP with TGP improved membrane protein expression in *E. coli*.** By now we had demonstrated TGP as an improved reporter for FSEC-TS at high temperatures. Enabling high-level expression of POI is another characteristic of an ideal fusion partner. Recently, it has been reported[66] that the yield of membrane proteins expression in mammalian cells can vary by up to 5 folds when fused with different GFP variants. As part of the characterization, the effect of TGP on membrane protein expression was also compared to GFPs in four expression systems.

Three different methods were used for assessment: fluorescence counts of cell culture, in-gel fluorescence of cell lysate or membranes, and FSEC of solubilized fluorescent fusion proteins. The fluorescence counting method is quick but risks over-estimation because the calculation does not exclude free GFP. In-gel fluorescence reveals the integrity of fusion proteins but is less accurate because the densitometry analysis is only semi-quantitative with a relatively narrow linear range. Besides, it gives little information for membrane protein folding because of the presence of denaturing detergent SDS. The FSEC method, although slightly more time-consuming, can provide more relevant information about membrane protein quality. Owing to the chromatographic separation, the profile reflects mono-dispersity and degradation severity. Therefore, we introduce Yield$_{FSEC}$ as a relative term to compare the effect of fluorescent protein on meaningful membrane protein expression levels (See Methods). To this end, the peak density corrected by their respective fluorescence-concentration relationship was used for comparison. Because only the fraction that was soluble in the mild detergents and eluted at a meaningful internal pore volume was used, it should provide both quantitative and qualitative information about the fusing membrane proteins.

We first ran the test for the *E. coli* system using *aa*PlsY and *ms*SI. Based on fluorescence counting, replacing ecGFP

with TGP increased the expression level to 2.5 folds for *aa*PlsY (Fig. 3a) and 5 folds for *ms*SI (Fig. 3b). In both cases, in-gel fluorescence analysis showed a major band corresponding to the molecular weight of the fusion protein (Fig. 3c, d, Supplementary Fig. 1b, 1c), and densitometry analysis showed a ~1.5-fold expression level with TGP over ecGFP (Fig. 3c, d, Table 1). In addition, the FSEC intensity of the TGP fusion was higher than the ecGFP fusion: 1.8-fold for *aa*PlsY (Figs. 3e) and 2.5-fold for *ms*SI (Fig. 3f), corresponding to 1.7-fold and 2.3-fold in Yield$_{FSEC}$ (Table 1), respectively. Importantly, the FSEC profiles were superimposable between the two fluorescent tags (Fig. 3g, h), suggesting tag type had no impact on the gel permeation behavior for both membrane proteins.

**Replacing GFP with TGP improved membrane protein expression in *S. cerevisiae*.** The test for the *S. cerevisiae* system was performed with *h*SI and PIS. Although no notable differences were observed between the TGP- and scGFP-fusion proteins based on fluorescence counting (Fig. 4a, b), in-gel fluorescence (Fig. 4c, d, Supplementary Fig. 1d, 1e) and Yield$_{FSEC}$ analysis (Fig. 4e, f, Table 1) showed a 2-fold expression level with TGP compared to scGFP for both membrane proteins. Similar to the *E. coli* system, the FSEC retention profile was unaffected by tag type for both *h*SI and PIS (Fig. 4g, h).

**Replacing GFP with TGP improved membrane protein expression in insect cells.** When tested in insect cells, replacing scGFP with TGP increased the expression of *h*SI (Fig. 5a–d) by 2 folds, based on the in-gel fluorescence (Fig. 5b, Supplementary Fig. 1f) and Yield$_{FSEC}$ analysis (Fig. 5c, Table 1). This trend was also observed for PIS, which showed a 1-fold increase when replacing scGFP with TGP (Fig. 5e–h, Supplementary Fig. 1g, Table 1). Again, the FSEC profile of the membrane proteins were the same regardless of the fluorescent tags, although more fractional degradation was observed for the TGP-tagged membrane proteins (Fig. 5d, h).

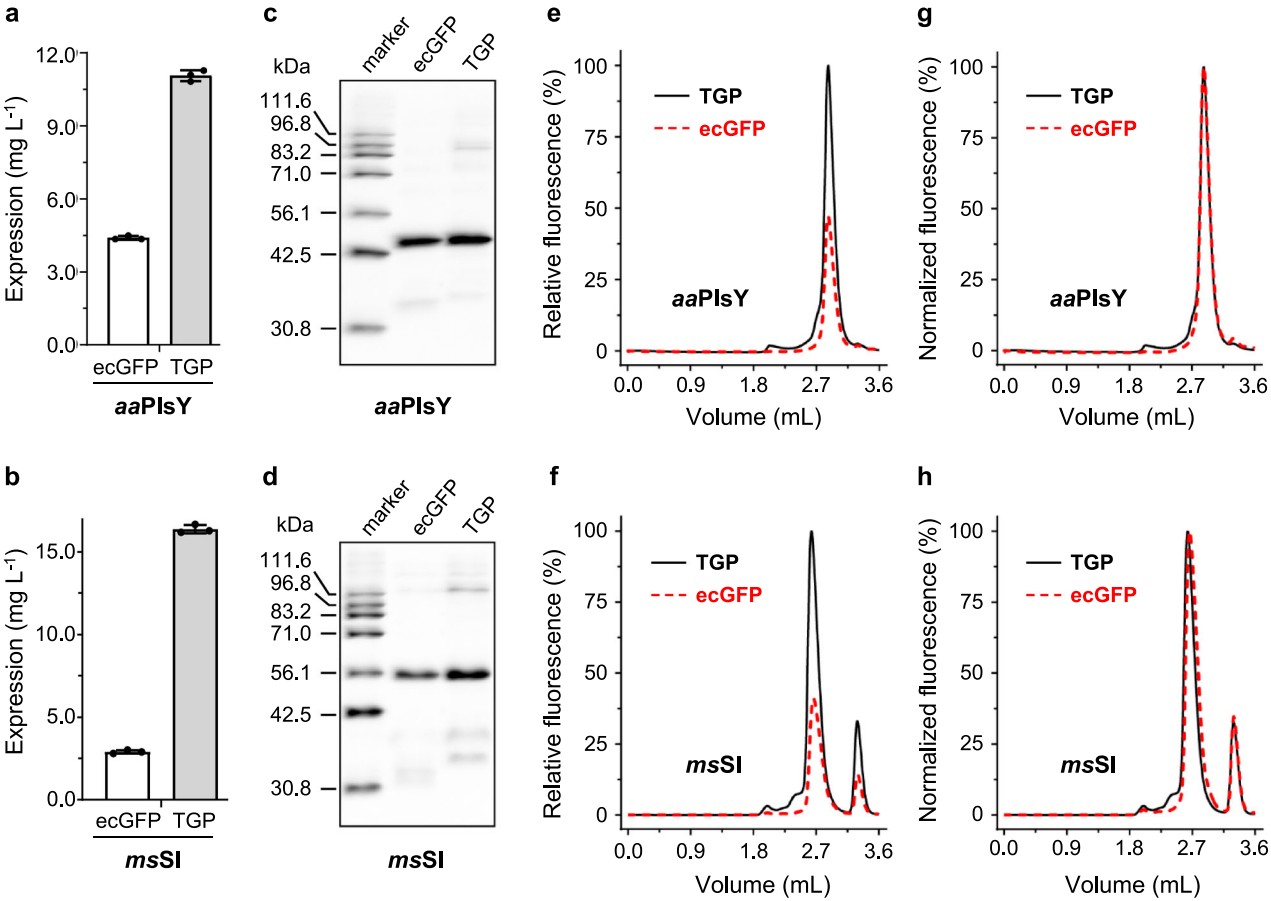

**Fig. 3 Replacing GFP with TGP improved expression of *aa*PlsY and *ms*SI in *E. coli*.** Assessment of expression level was based on fluorescence counts (**a**, **b**), relative in-gel fluorescence (**c**, **d**), and relative FSEC intensity (**e**, **f**). Normalized FSEC traces are shown in **g**, **h**. Mean and standard deviation (**a**, **b**) or a representative (**c–f**) of three independent experiments using different colonies are shown. In **e–h**, $V_o$ was 2.02 mL and $V_t$ was 4.55 mL. To save column and reagents, most FSEC assays in this report were run for 3.60 mL because fluorescence signal after the complete elution of free fluorescent proteins (~3.50 mL) was negligible in test runs. In **e**, **g**, **f**, and **h**, TGP traces are shown as black solid lines and ecGFP traces are shown as red dashed lines. FSEC fluorescence-detection size exclusion chromatography, GFP green fluorescent protein, TGP thermostable GFP, $V_o$ void volume, $V_t$ total volume.

**Replacing GFP with TGP improved membrane protein expression in mammalian cells.** TGP showed remarkable improvement for *h*SI expression in the mammalian system. First, for maximal transient expression level, lower amount of plasmid was required for *h*SI-TGP (2 mg L$^{-1}$) than for *h*SI-scGFP (3 mg L$^{-1}$). When assessed under optimal conditions, the yield for *h*SI-TGP was over 10 folds compared with *h*SI-scGFP using the three aforementioned quantitative methods (Fig. 6a–d, Supplementary Fig. 1h, Table 1). The low level for *h*SI-scGFP (1.0 mg L$^{-1}$) could mean a challenging project for most crystallography groups because of high costs with the large-scale mammalian expression for membrane protein production in milligram quantities. By contrast, that for *h*SI-TGP (11.1 mg L$^{-1}$) would make a promising structural project.

Notably, the FSEC signal for *h*SI-scGFP was invisibly flat when plotted in scale with *h*SI-TGP (Fig. 6c) due to the combinational effect from weak expression and low brightness associated with scGFP. The difference in peak intensity between the two was 38 folds, translating to a ~13-fold difference in Yield$_{FSEC}$, which was in close agreement with the other two methods. Despite the yield differences, the two *h*SI forms displayed nearly superimposable FSEC profiles (Fig. 6d), again suggesting a similar overall fold.

Higher expression level with TGP in comparison to scGFP was also observed for PIS, although not as dramatic as for *h*SI. As judged by fluorescence counting, switching from scGFP to TGP

increased the expression level of PIS to 3.8 folds (Fig. 6e). In-gel fluorescence (Fig. 6f, Supplementary Fig. 1i) and FSEC results (Fig. 6g) confirmed this trend, displaying a 2-fold difference (Table 1). Despite the overall superimposable profiles (Fig. 6h), the free TGP peak was much more evident than scGFP, a trend also seen in in-gel fluorescence. Quantification of the peaks showed that a quarter of TGP-fusion protein was degraded, which means the gap between the expression yields could have been larger had the degradation be controlled.

**TGP sybodies enabled affinity purification of membrane proteins.** As part of the GFP toolkit, single-chain camelid antibodies (nanobodies)[67,68] and designed ankyrin repeat proteins (DARPins)[69] are available for affinity chromatography, contributing to the purification of several high-profile membrane proteins for structure determination[70–75]. To develop equivalents for TGP, we selected synthetic nanobodies (sybodies) from a recently established library[59] using a strategy that combines ribosome display and phage display. Since the concave sybody library[59] was randomized on the basis of a GFP nanobody known as the enhancer (PDB 3K1K), which bound sideways and contains a short complementarity determining regions 3 (CDR3), this library was used for performing the selections. After three rounds of panning with successively lower TGP concentrations, libraries with enriched binders were subcloned from the phage display system to expression plasmids for screening at a single-colony level.

**Table 1 Comparison of TGP with GFPs for FSEC-TS assay and membrane protein expression.**

| Expression host | Protein | Fusion | App. $T_m$ (°C)[a] | Yield (mg L$^{-1}$)[b] | In-gel fluorescence | | FSEC | |
|---|---|---|---|---|---|---|---|---|
| | | | | | Intst. (AU) | Yield (%)[c] | Intst. (%) | Yield (%)[d] |
| *E. coli* | *aa*PlsY | TGP | 91.1 | 11.1 ± 0.2 | 100 ± 15.2 | 100 ± 15.2 | 100 ± 8.4 | 100 ± 8.4 |
| | | ecGFP | 73.5 | 4.4 ± 0.1 | 60.6 ± 14.3 | 61.2 ± 14.5 | 55.1 ± 6.7 | 60.5 ± 6.1 |
| | *ms*SI | TGP | 33.3 | 16.4 ± 0.3 | 100 ± 17.0 | 100 ± 17.0 | 100 ± 15.3 | 100 ± 15.3 |
| | | ecGFP | 32.0 | 2.9 ± 0.1 | 56.9 ± 7.9 | 57.5 ± 8.0 | 39.2 ± 3.6 | 43.1 ± 4.0 |
| *S. cerevisiae* | *h*SI | TGP | 49.4 | 1.7 ± 0.1 | 100 ± 14.5 | 100 ± 14.5 | 100 ± 4.6 | 100 ± 4.6 |
| | | scGFP | 44.3 | 1.6 ± 0.1 | 21.3 ± 6.7 | 46.3 ± 14.6 | 14.3 ± 0.7 | 43.3 ± 2.1 |
| | PIS | TGP | 41.3 | 1.5 ± 0.1 | 100 ± 40.3 | 100 ± 40.3 | 100 ± 29.1 | 100 ± 29.1 |
| | | scGFP | 40.5 | 1.0 ± 0.0 | 21.4 ± 4.6 | 46.6 ± 10.0 | 14.1 ± 0.8 | 42.7 ± 2.4 |
| *Sf9* | *h*SI | TGP | / | 12.5 ± 0.7 | 100 ± 21.5 | 100 ± 21.5 | 100 ± 1.0 | 100 ± 1.0 |
| | | scGFP | / | 6.0 ± 0.2 | 17.4 ± 2.5 | 37.8 ± 5.4 | 10.9 ± 0.8 | 33.0 ± 2.4 |
| | PIS | TGP | / | 12.5 ± 0.3 | 100 ± 24.2 | 100 ± 24.2 | 100 ± 11.2 | 100 ± 11.2 |
| | | scGFP | / | 7.0 ± 0.2 | 24.7 ± 3.5 | 53.6 ± 7.5 | 15.8 ± 1.0 | 47.8 ± 3.2 |
| Expi293 | *h*SI | TGP | / | 11.1 ± 0.8 | 100 ± 12.7 | 100 ± 12.7 | 100 ± 5.4 | 100 ± 5.4 |
| | | scGFP | / | 1.0 ± 0.1 | 3.0 ± 0.5 | 6.5 ± 1.1 | 2.6 ± 0.1 | 7.8 ± 0.4 |
| | PIS | TGP | / | 2.3 ± 0.5 | 100 ± 28.3 | 100 ± 28.3 | 100 ± 18.3 | 100 ± 18.3 |
| | | scGFP | / | 0.6 ± 0.1 | 22.0 ± 1.6 | 47.7 ± 3.4 | 14.2 ± 1.0 | 43.0 ± 3.0 |

[a]Apparent $T_m$ was calculated by sigmoidal dose-response fitting of data from single experiments.
[b]Yield, as mean ± standard deviation, was calculated based on the fluorescence counts of cells from three independent experiments on different colonies for bacterial and yeast expression, and on cells with different passage numbers for baculovirus and mammalian expression.
[c]Percent yield relative to membrane protein-TGP was calculated based on integrated arbitrary intensity (intst, AU) corrected with calibrated quantity-intensity correlation. Results (mean ± standard deviation) are from three independent experiments as in [b].
[d]Percent yield relative to membrane protein-TGP was calculated using intensity corrected with calibrated quantity-fluorescence correlation. Data (mean ± standard deviation) are from three independent experiments as in [b].

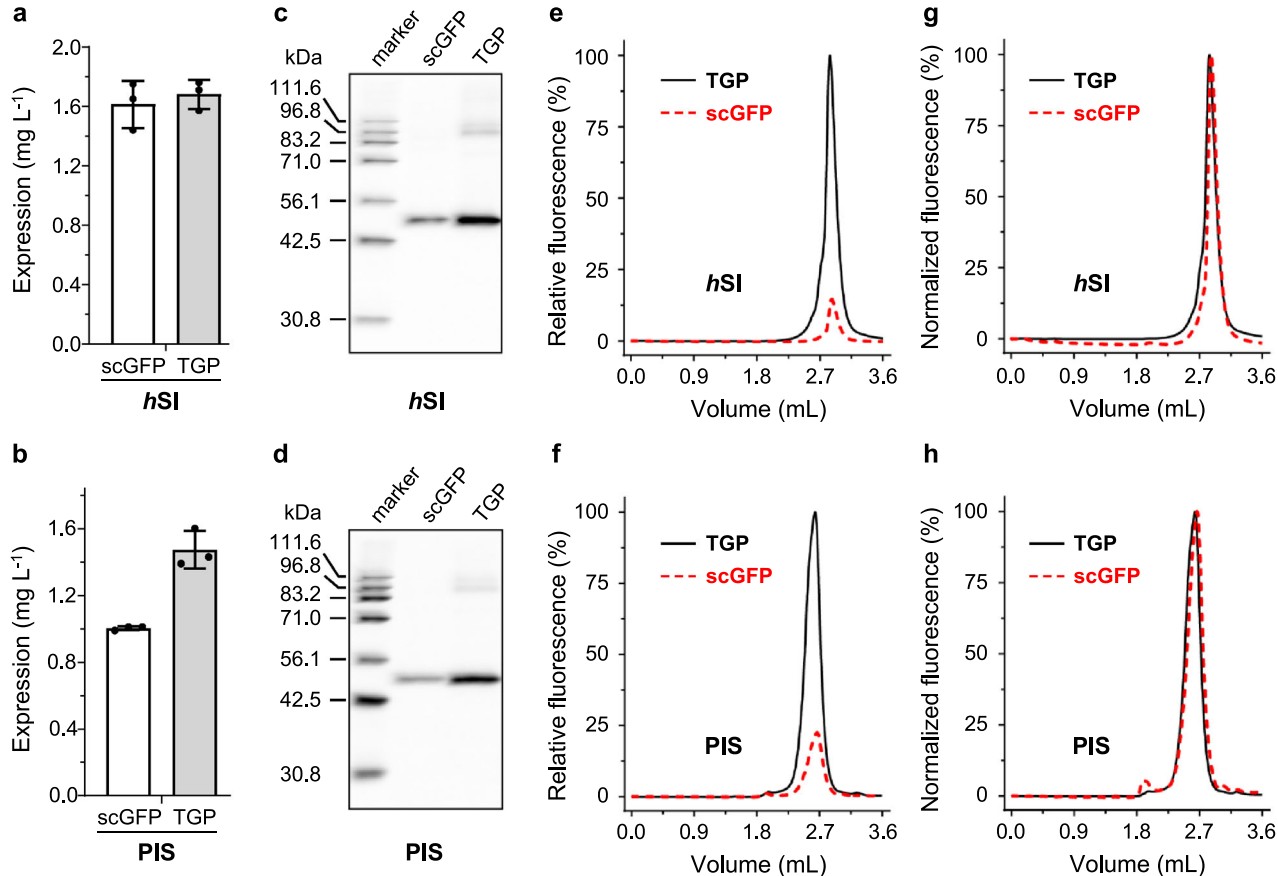

**Fig. 4 Replacing GFP with TGP improved expression of two human membrane proteins in *S. cerevisiae*.** Assessment of expression level was based on fluorescence counts (**a**, **b**), in-gel fluorescence (**c**, **d**), and relative FSEC intensities (**e**, **f**). Normalized FSEC traces are shown in **g**, **h**. Mean and standard deviation (**a**, **b**) or a representative (**c**–**f**) of three independent experiments on different colonies are shown. In **e**, **g**, **f**, and **h**, TGP traces are shown as black solid lines and scGFP traces are shown as red dashed lines. FSEC fluorescence-detection size exclusion chromatography, GFP green fluorescent protein, TGP thermostable GFP.

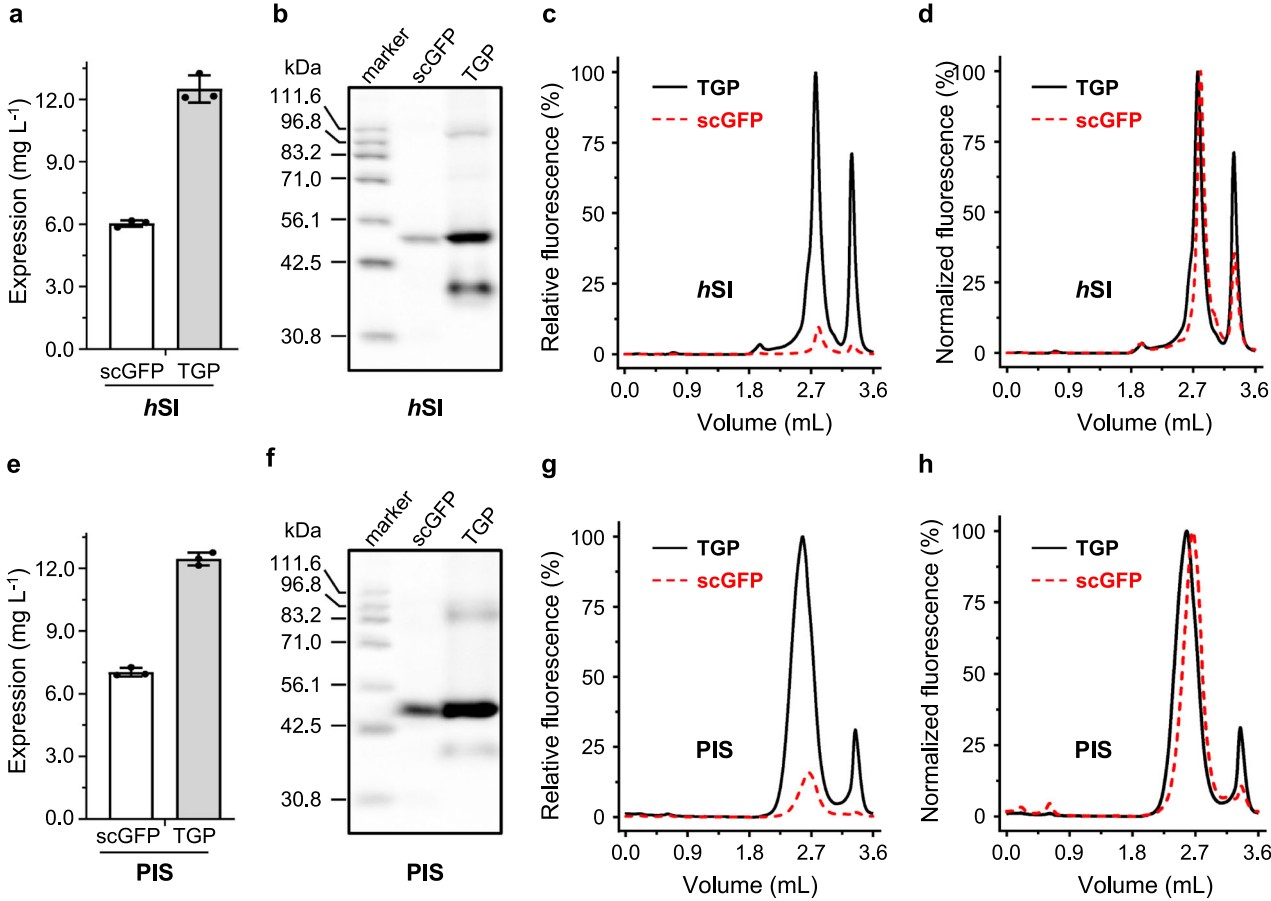

**Fig. 5 Replacing GFP with TGP improved expression of two human membrane proteins in insect cells.** Assessment of *h*SI expression was based on fluorescence counts (**a**), in-gel fluorescence (**b**), and relative FSEC intensity (**c**). Normalized FSEC traces of *h*SI-TGP and *h*SI-scGFP are shown in **d**. Assessment of PIS expression was based on fluorescence counts (**e**), in-gel fluorescence (**f**), and relative FSEC intensity (**g**). Normalized FSEC traces of PIS-TGP and PIS-scGFP are shown in **h**. Mean and standard deviation (**a**, **e**) or a representative (**b-d**, **f-h**) of three independent experiments on cells with different passage numbers are shown. In **c**, **d**, **g**, and **h**, TGP traces are shown as black solid lines and scGFP traces are shown as red dashed lines. FSEC fluorescence-detection size exclusion chromatography, GFP green fluorescent protein, TGP thermostable GFP.

Prescreening of 188 colonies using a pull-down assay (see Methods) identified 86 hits that showed higher TGP fluorescence in the elution compared to the negative control. Further screening for earlier retention volumes of TGP FSEC upon incubation with periplasmic extracts identified four binders (Fig. 7a, Supplementary Fig. 5). TGP-sybody complexes also formed in preparative size exclusion chromatography (Fig. 7b, Supplementary Fig. 1j). Biolayer interferometry assays showed that the sybodies bound to TGP with affinities in the range of 3.3–10.4 nM (Fig. 7c–f). The sybodies could all be purified from *E. coli* with relatively high yield ranging from 10–48 mg L$^{-1}$.

The feasibility of using the sybodies to purify TGP-fusion membrane proteins was demonstrated using *ms*SI-TGP expressed in *E. coli*. To this end, purified sybody 44 (Sb44) was conjugated to a solid support by amine coupling. *ms*SI-TGP was then purified by standard affinity chromatography protocols, and TGP-free *ms*SI was released from the affinity beads by enzymatic digestion at a pre-engineered 3C protease site. The purity of *ms*SI was comparable to that purified using immobilized metal affinity chromatography (Fig. 7g, Supplementary Fig. 1k, 1l). In practice, immobilized metal affinity chromatography and nanobody-affinity chromatography could be performed in tandem to increase purity especially for those expressed at low abundance.

Nanobodies recognizing different parts of GFPs have been reported in literature[76]. Such nanobodies could be used to construct biparatopic fusion to enhance affinity[69,77]. To seek such nanobodies for TGP, we performed FSEC analysis of the Sb66-TGP complex in the presence of other binders. This identified Sb92 as a noncompeting partner for Sb66 for TGP-binding (Fig. 7h).

**Crystal structure of the TGP-Sb44 complex.** For further characterization, we obtained crystals of TGP-Sb44 (Supplementary Fig. 6) which diffracted to 2.0 Å at the synchrotron. The structure of the complex (Fig. 8, Table 2) was solved by molecular replacement using existing TGP and nanobody structures[58,59,78] as search models. The asymmetric unit contained two TGP-Sb44 heterodimers, which were highly similar (Cα root mean square deviation of 0.29 Å). We focused on one TGP-Sb44 dimer (Chain IDs C and D) for structural analysis.

The Cα root mean square deviation between the previous TGP-alone structure (PDB 4TZA[58]) and the Sb44-bound form was 0.20 Å. Thus, the binding did not cause noticeable conformation change of TGP, reflecting the ultrastability and thus structure rigidity of TGP. Consistent with this, unlike those reported for GFPs[68], none of the sybodies changed the fluorescent property of TGP.

Sb44 binds to one side of the TGP barrel (Fig. 8a) with the surfaces showing electrostatic complementarity (Fig. 8b).

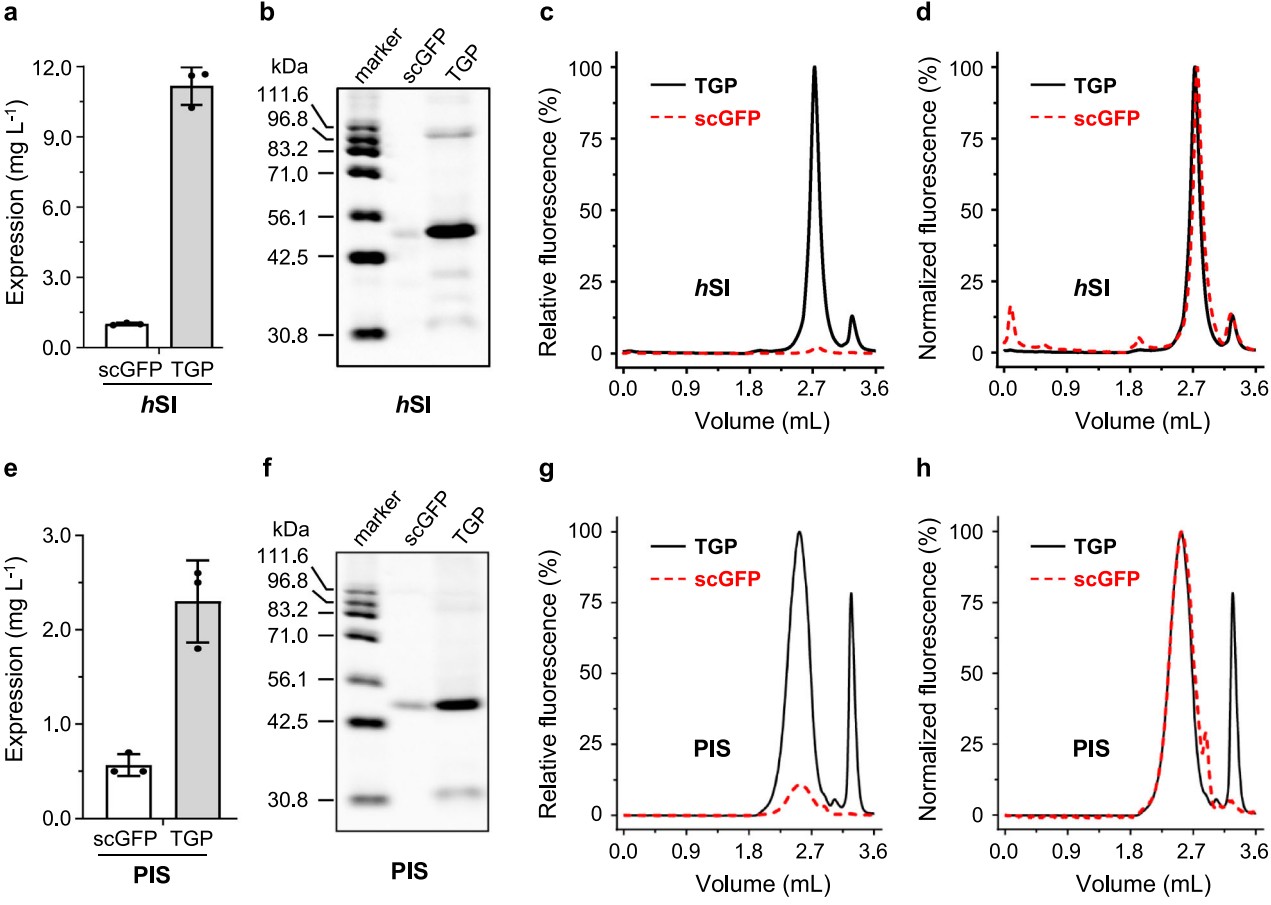

**Fig. 6 Replacing GFP with TGP improved expression of two human membrane proteins in mammalian cells.** Assessment of *h*SI expression was based on fluorescence counts (**a**), in-gel fluorescence (**b**), and relative FSEC intensity (**c**). Normalized FSEC traces of *h*SI-TGP and *h*SI-scGFP are shown in **d**. Assessment of PIS expression was based on fluorescence counts (**e**), in-gel fluorescence (**f**), and relative FSEC intensity (**g**). Normalized FSEC traces of PIS-TGP and PIS-scGFP are shown in **h**. Mean and standard deviation (**a**, **e**) or a representative (**b**–**d**, **f**–**h**) of three independent experiments on cells with different passage numbers are shown. In **c**, **d**, **g**, and **h**, TGP traces are shown as black solid lines and scGFP traces are shown as red dashed lines. FSEC fluorescence-detection size exclusion chromatography, GFP green fluorescent protein, TGP thermostable GFP.

Interestingly, although Sb44 binds at approximately the opposite side of the barrel compared to the GFP nanobody enhancer (PDB 3K1K) (Supplementary Fig. 7a), its paratope surface is also concave-shaped (Supplementary Fig. 7b–d) as anticipated by the design of the concave sybody library[59]. A PISA[79] analysis showed an interface (835.6 Å$^2$) that was larger than those for the GFP nanobodies[68] (enhancer, 672 Å$^2$; minimizer, 652 Å$^2$). As typically observed for nanobodies, all three CDRs were involved in the recognition of the epitope (Fig. 8c–e, Supplementary Fig. 8, Supplementary Fig. 9) consisting of TGP residues from the β-strand 2, 4, 5, 6, and 9, which are 3-dimensionally close but primarily distant. Similar to the GFP enhancer nanobody, several scaffold (non-CDR) residues of Sb44 also participated in the binding. In detail, the interaction was mediated by two salt bridges, nine hydrogen bonds, and a rich network of hydrophobic interactions made of apolar residues and the hydrocarbon part of several hydrophilic side chains (Fig. 8d, e). Peculiarly, six of the nine hydrogen bonds involved backbone amino and carbonyl groups.

## Discussion

FSEC-TS assays for fluorescently tagged membrane proteins are best performed at temperatures where the tags are largely unaffected (~25% drop of fluorescence) so that the decrease of fluorescence can be largely attributed to the aggregation/precipitation of protein of interest (POI). We recommend the

upper limit for FSEC-TS assay as ~90 °C for TGP, ~74 °C for ecGFP, and ~64 °C for scGFP in the presence of dodecylmaltoside. Therefore, TGP broadens the FSEC-TS temperatures drastically.

Previously, FSEC-TS assays at high temperatures (>76 °C) rely on tryptophan fluorescence of pure proteins, which, for membrane proteins, can be difficult to obtain[46]. Alternatively, they can be performed with the modestly-stable GFPs in harsher detergents, typically those with shorter chains such as decylmaltoside and octylglucoside. Accordingly, the stability of membrane protein, but probably not of GFPs, drops, allowing $T_m$ measurement under comprising conditions for membrane proteins. While arguably straightforward, such experiments, bring disadvantages. First, the POI may not be stable at all in harsh detergents. Second, high concentrations need to be used for shorter-chain detergents because of their high critical-micelle-concentrations. This could increase the cost of FSEC-TS by one to two orders of magnitude. Third, because purifications are usually performed in mildest detergents possible, the results obtained with harsh detergents would be less informative to guide the design of purification strategies. The use of TGP avoids such problems.

The FSEC-TS trace increases before dropping for some TGP-fusion proteins (Fig. 2b, e, f), but not for all (Fig. 2a, c, d), a phenomenon also seen in literature[43,46,64]. Because the increase was not observed for the free TGP, and because it appeared at different temperatures for different membrane proteins, we

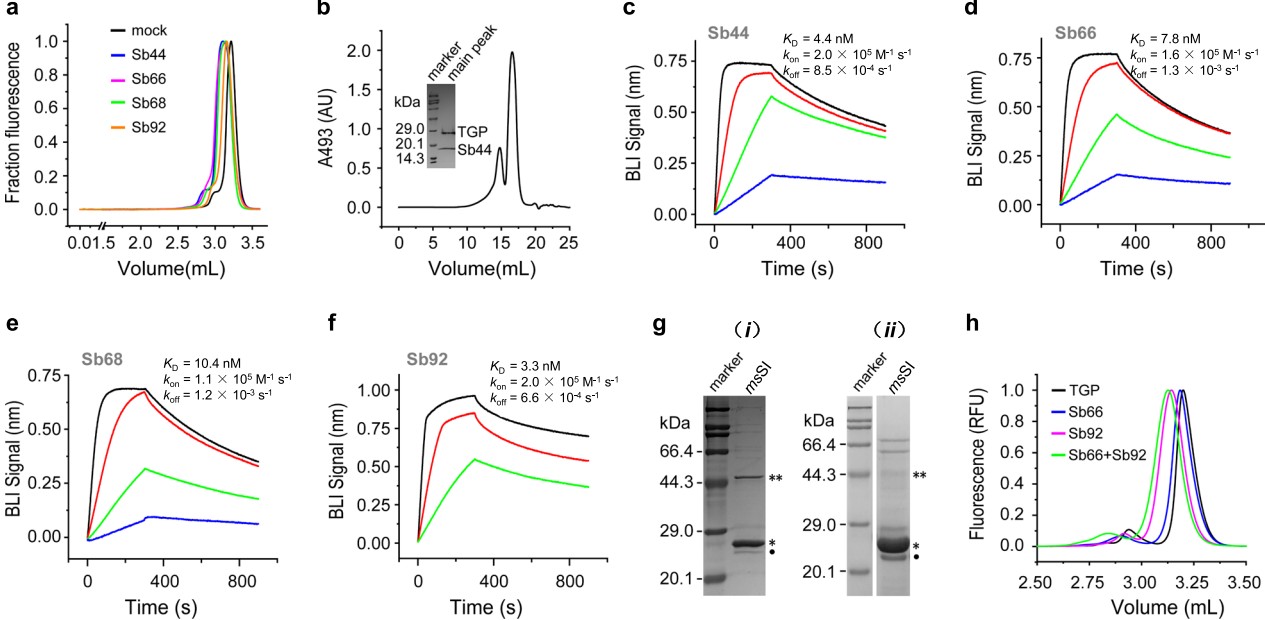

**Fig. 7 Characterization of TGP-binding sybodies. a** Identification of TGP binders using FSEC. Four TGP binders (Sb44, blue; Sb66, magenta; Sb68, green; and Sb92, orange), but not a binder for the unrelated maltose-binding protein[59] (mock, black), caused earlier elution of TGP. **b** SEC of TGP-Sb44 with absorbance detection at 493 nm. Inset shows the SDS-PAGE of the main-peak fraction. **c–f** Biolayer interferometry (BLI) assay for binding kinetics with TGP immobilized and sybodies as analyte at 7.4 (blue), 22 (green), 66 (red), and 200 nM (black) concentrations. **g** Affinity purification of *ms*SI using conjugated Sb44 (***i***) and immobilized metal affinity chromatography (***ii***). An asterisk indicates *ms*SI at 27.1 kDa; A double asterisk indicates possible *ms*SI dimer. The apparent molecular weight of *ms*SI was slightly smaller than the theoretical value (30.0 kDa), a phenomenon frequently observed for hydrophobic membrane proteins and other sterol isomerase homologs[14,64,91]. The faint band below *ms*SI (dot) was residual 3C protease. Other bands were probably contaminants. **h** FSEC of TGP in the absence (black) or in the presence of Sb66 (blue), Sb92 (magenta), or both sybodies (green). FSEC fluorescence-detection size exclusion chromatography, GFP green fluorescent protein, sybody synthetic nanobody, TGP thermostable GFP.

propose it is not intrinsic for TGP. Instead, it probably reflects the folding states of POI. It has been reported that GFP fluorescence can respond to subtle environmental differences caused by POI unfolding (before aggregation), and the resulted changes in intensity (decrease in those cases) can be used directly to monitor unfolding by differential scanning fluorimetry[80]. In our assays, the rise-phase, appearing just before the drop-phase (aggregation), might also reflect local denaturation of POIs before aggregation.

The $T_m$ values measured with GFP and TGP for the same POI were mostly consistent. However, a 5-°C increase in apparent $T_m$ was observed for *h*SI-TGP compared with *h*SI-scGFP. Because *h*SI and the fluorescent proteins are unrelated and exist as distinct domains, it is very unlikely that the stability of *h*SI was intrinsically enhanced by fusion to a more stable partner (TGP); rather, like a stable chaperon, TGP may have prevented the fusion protein as a whole from precipitating during heating, thus showing a higher apparent $T_m$, or in essence, $T_{aggregation}$.

We note a 18-°C decrease of the apparent $T_m$ for *h*SI-TGP after purification (Fig. 2d, 31.2 °C) compared with that obtained using the membrane solubilization fraction (Fig. 2c, 49.4 °C). This is probably caused by the likely delipidation events during chromatography steps. The loss of stability after purification was also observed for PIS, in which case the PIS-TGP fusion protein precipitated during our attempts in isolating pure TGP-fused and TGP-free PIS for stability comparison as was done for *tt*SI and *h*SI (Fig. 2d).

As an indirect assay, how apparent $T_m$ values would differ from actual $T_m$ values (as determined by functional assays) is always a topic of general interest. It is widely accepted that such apparent $T_m$ values should be treated with caution. Only when denaturation coincides with aggregation would the values be the same.

Occasionally, membrane proteins, even for those with a high apparent $T_m$ in the fusion form, precipitate upon the removal of the fluorescent protein, suggesting protective effect of these stable tags on membrane proteins[27,81]. For crystallization, it may require changing to fluorescent tag-free constructs so that the POI did not experience compositional shock during later purification steps. Despite these limitations, FSEC-TS should be considered when absolute $T_m$ values are not necessarily required, for example, in the screening of stable homologs or mutants for structural study, or screening of lipids, ligands for purification, owing to its simplicity, general applicability, and high-throughput feature.

Interestingly, the $T_m$ of *sp*PlsY measured by FSEC-TS only differed slightly from that measured by enzymatic assays. Probably, as a relatively small protein without multi-domains, its unfolding followed a simple path, and it remained active until heat-induced aggregation.

All the tested membrane proteins displayed higher expression levels with TGP compared to GFPs in both prokaryotic and eukaryotic systems. For *h*SI, substituting GFP with TGP enhanced the expression level by a remarkable 10 folds when expressed in mammalian cells. The mechanism for this remains to be studied. Possibly, the fusion with a rock-stable partner helps the membrane proteins escape the surveillance system for cellular degradation.

The increase of POI expression by the replacement of TGP for GFP should not be interpreted such that the TGP tag always increases expression of POIs in comparison with their tag-free forms. Including a chaperon may be beneficial for heterologous expression, but this will inevitably increase energy expenditure of the host. In addition, depending on the POIs, tags may interfere with POI's folding and localization, and hence may influence

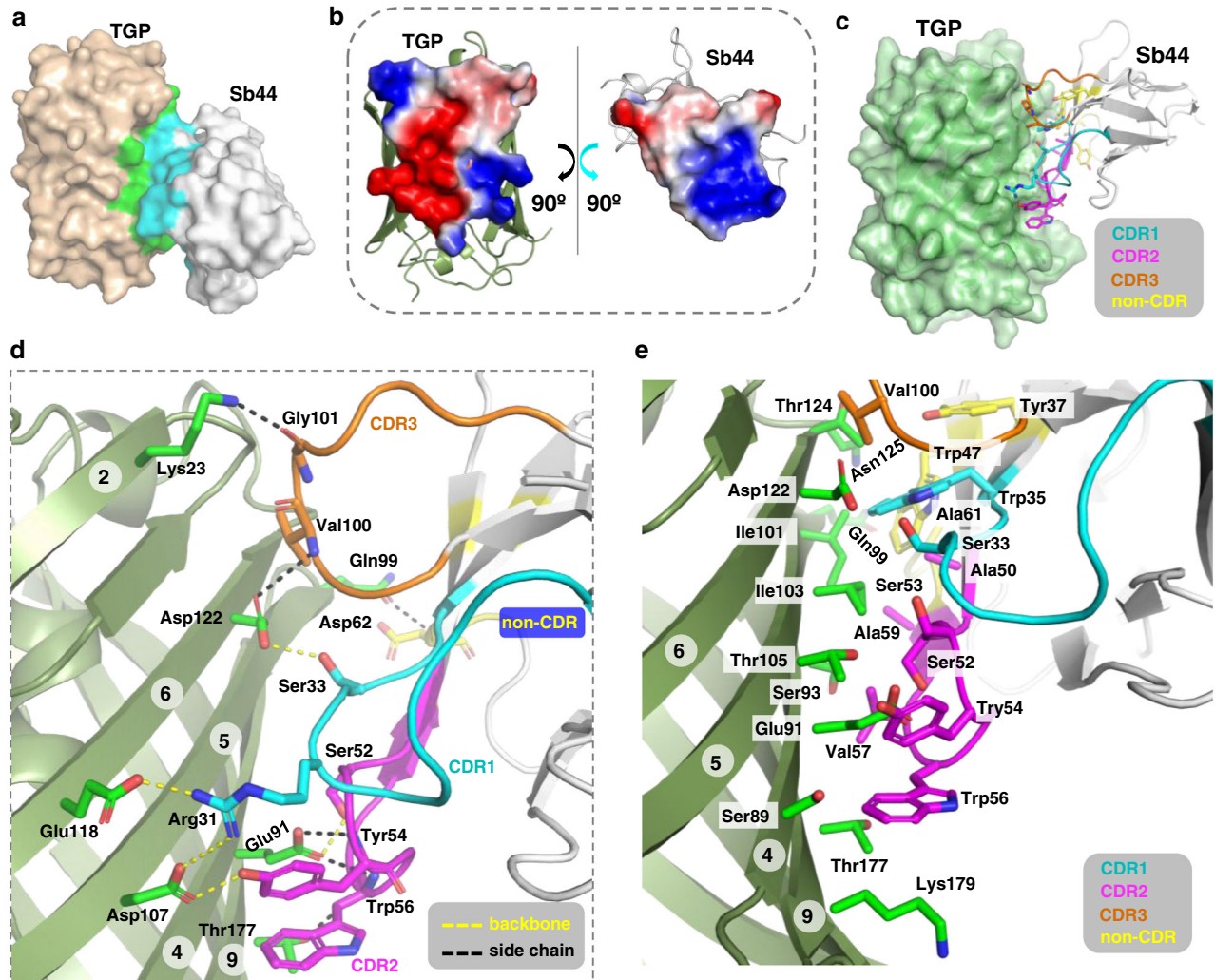

**Fig. 8 Crystal structure of the TGP-Sb44 complex. a** Surface representation with interface residues colored green (TGP) and cyan (Sb44). **b** 'Open-book' representation of the electrostatic potential molecular surface generated using the Adaptive Poisson-Boltzmann Solver[79] module in PyMol[90]. **c** Sb44 binds with its concave surface to TGP. Interacting non-CDR and CDRs residues are shown as sticks with carbon atoms in indicated colors. **d** Salt bridges and H-bonds between TGP and Sb44 with a cutoff distance of 3.9 Å. Yellow dash lines (5) indicate interactions between side chains. Black dash lines (6) indicate H-bonds that partly contributed by backbone atoms. **e** Hydrophobic interaction network formed by hydrophobic residues or the apolar part of hydrophilic residues. Circled numbers indicate TGP β-strands. CDR complementarity-determing region, TGP thermostable green fluorescent protein, sybody synthetic nanobody.

negatively on production yield[14]. Therefore, the effect of fluorescent protein tags on membrane protein expression should be tested on a case-by-case basis, when needed.

We note that, although revealing the same trend in expression level, the fluorescence counting, in-gel fluorescence, and Yield$_{FSEC}$ methods sometimes gave inconsistent results. This is partly caused by the fact that these assays used different fractions. In some cases, the in-gel fluorescence and Yield$_{FSEC}$ experiments used membranes removal of free fluorescent proteins, while the fluorescence counting method did not exclude them. Overall, for comparison, we credit the Yield$_{FSEC}$ more than the other two methods due to reasons mentioned in the Results section, and we use this now routinely in the lab for the optimization of expression conditions.

While the generality of high expression remains to be tested, the gained sensitivity due to TGP's high brightness should be beneficial for fluorescence-based assays, especially with membrane proteins that express poorly in costly eukaryotic or in vitro systems. This is especially true in comparison to scGFP. For

example, to evaluate the yield of membrane protein-GFPs, enough signal, usually 10 times over the background of medium (150–280 relative fluorescence units depending on the medium types) are needed. This corresponds to 6–11 mg L$^{-1}$ of expression level in the case of scGFP. Since membrane protein expression levels are usually below this, cells are concentrated by centrifugation and resuspension before fluorescence measurement to enhance the signal-to-noise ratio. The use of TGP should enable such measurements with less cell culture or costly cell-free reagents, by a factor of 2, when at a same expression level. Similarly, the loading for TGP-based FSEC can be reduced by a factor of 3, lowing the risk of clogging expensive columns.

Although the TGP form here has been monomerized, signs of dimer were observed in in-gel fluorescence assays (Fig. 1b) even with fused membrane proteins (Figs. 3c, d, 4c, d, 5b, f, 6b, f). In line with this, TGP showed a peak at 3.02 mL, consistent with a dimer based on calibrated molecular weight. Interestingly, the putative dimer resisted high temperatures up to 80 °C (Supplementary Fig. 2a).

**Table 2 Data collection and refinement statistics.**

| | TGP-Sb44 |
|---|---|
| **Data collection** | |
| Space group | $P2_12_12_1$ |
| Cell dimensions | |
| $a, b, c$ (Å) | 51.61, 83.49, 184.56 |
| $\alpha, \beta, \gamma$ (°) | 90, 90, 90 |
| Wavelength (Å) | 0.97930 |
| Resolution (Å) | 49.53–2.03 (20.9–2.03)[a] |
| $R_{merge}$ | 0.127 (1.694) |
| $R_{pim}$ | 0.037 (0.504) |
| $I/\sigma I$ | 13.9 (1.6) |
| Completeness (%) | 99.3 (91.3) |
| Multiplicity | 12.8 (11.9) |
| $CC*$[b] | 0.999 (0.896) |
| **Refinement** | |
| Resolution (Å) | 45.04–2.03 |
| No. reflections | 51,724 |
| $R_{work}/R_{free}$ | 0.1753/0.2155 |
| No. atoms | 5990 |
| Protein | 5435 |
| Ligand/ion | 86 |
| Water | 469 |
| No. residues | 670 |
| B-factors (Å²) | 39.40 |
| Protein | 39.07 |
| Ligand/ion | 36.65 |
| Water | 43.78 |
| R.m.s deviations | |
| Bond lengths (Å) | 0.007 |
| Bond angles (°) | 0.90 |
| Ramachandran | |
| Favored (%) | 98.48 |
| Allowed (%) | 1.37 |
| Outlier (%) | 0.15 |
| **PDB ID** | 6LZ2 |

[a]Highest resolution shell is shown in parenthesis.

[b]$CC* = \sqrt{\frac{2CC_{1/2}}{1+CC_{1/2}}}$

The previous crystal structure contains four TGP protomers (Supplementary Fig. 10a) in the asymmetric unit[58] while our structure contains two protomers that are inverted (Supplementary Fig. 10b). Comparing the structures revealed that the surface involved in protomer interactions in the previous structure[58] (PDB 4TZA) overlaps with that in our structure (Supplementary Fig. 10c, 10d). Because the two interacting protomers from the two structures ($P_C$ and $P_{C'}$) had different orientations (Supplementary Fig. 10c), the interface may be a result of nonspecific crystal artifacts. Nevertheless, this common interface should be considered for mutation should further TGP monomerization be needed. Interestingly, the other major interaction surfaces for protomers in the previous structure was exploited by Sb44 for binding (Supplementary Fig. 10e, 10f).

In summary, the work here presents encouraging results for TGP as a replacement of current GFP tags for membrane protein production, purification, and stability measurement.

## Methods

**Molecular cloning**. The coding sequence of TGP (PDB entry 4TZA, originally named as 'Azami-Green')[56,58] was synthesized by overlapping PCR using oligos that are typically 59-bp long with 19–20 bp of annealing region. The gene encoding *ms*SI (NCBI WP_058126696.1) was amplified from the genomic DNA of *M. segamatis* using primer pairs 5′- GTCGACGGGCCCGGGATCCACATCCGACA TCGCGACAC-3′ and 5′- GAATTCGAGCTCGGTACCCCGAGGGCCTGCCC-3′. The PCR product was digested with *BamH*I and *Kpn*I and ligated into the vector p3EG and pETSG. The plasmid for *h*SI (NCBI NP_006570.1) expression was

constructed previously[14]. The gene encoding PIS (NCBI NP_006310.1) was amplified from a cDNA clone (#BC001444) using primer pairs 5′-GGGAA-TATTtaaaaaATGtctggatccCCAGACGAAAATATCTTCCTG-3′ and 5′-gccctgaaa-cagcacttccagggtaccCTTCTTCTTGGCGCGGTCTGC-3′. The PCR product was digested with *BamH*I and *Kpn*I and ligated into vector p3YG and pYTSG. The fragments of *h*SI and PIS were also subcloned into vectors of p3FG and pFTSG for insect cell expression, and pMG and pBTSG for expression in mammal cells. Schematic maps of the vectors are shown in Supplementary Fig. 11.

**Protein purification—GFPs and TGP**. ecGFP was expressed in *E. coli* as a C-terminally His-tagged protein. BL21 (DE3) cells carrying the plasmid pEG[14] was induced at $OD_{600}$ of 0.6–0.8 for 16 h at 30 °C. Cells were suspended in a lysis buffer containing 150 mM NaCl, 50 mM Tris-HCl pH 8.0 and passed through a Constant cell disruptor (Cat. TS 0.75KW, Constant System, Daventry, UK) at 25 kpsi for 3 times. The lysate was heated at 65 °C for 10 min, cooled in ice-water, and clarified by centrifugation at $20,000 \times g$ for 30 min at 4 °C. The supernatant was incubated with 4 mL of Ni-NTA resin for 2 h with gentle agitation at 4 °C. The beads were loaded into a gravity column and washed with 20 column volume (CV) of lysis buffer supplemented with 50 mM imidazole. ecGFP was eluted using 250 mM imidazole in the lysis buffer.

scGFP was expressed in *Pichia pastoris* GS115 (lab collection, genotype *his4*) also as a C-terminally His-tagged protein. The strain was checked by its auxotrophic phenotype. Cells carrying the plasmid pZG[14] were cultured in 10 mL YPD medium in a 50-mL tube for culturing at 250 rpm at 30 °C. After 24 h, cells were seeded into fresh 1 L BMGY medium (2%(w/v) peptone, 1%(w/v) yeast extract, 1.34%(w/v) YNB, 0.4 mg $L^{-1}$ biotin, 1%(w/v) glycerol, 100 mM potassium phosphate pH 5.5). When $OD_{600}$ reached 15–20, cells were collected by centrifugation and resuspended with BMMY medium (replacing glycerol in BMGY with 0.5%(v/v) methanol) to $OD_{600}$ of 3–4 for protein expression. To compensate for metabolic consumption, methanol was added again to a 0.5%(v/v) after 24 h. Cells were harvested and resuspended in the same lysis buffer as for ecGFP, and lysed by passing through a Constant cell disruptor three times at 25, 28, and 30 kpsi. The purification was carried out as for ecGFP except that no heating was performed.

The expression and purification of TGP was performed using the same procedure as for ecGFP except that the lysate was heated at 80 °C instead of 65 °C before affinity chromatography.

The concentration of GFPs was determined using the molar extinction coefficient of 22,015 $M^{-1}$ $cm^{-1}$ with absorbance measured at 280 nm. The concentration of TGP was determined the same way with the molar extinction coefficient of 31,985 $M^{-1}$ $cm^{-1}$ at 280 nm.

**Protein purification—BirA (Uniprot P06709)**. *E. coli* BL21(DE3) cells carrying the plasmid pET-BirA (protein sequence: $His_6$-SSGLVPRGSH-BirA)[82] (a kind gift from Professor Ying Gao at the authors' institute) were induced with 0.1 mM IPTG for 15 h at 18 °C. Cell pellet from 1 L culture was resuspended with 80 mL lysis buffer containing 150 mM NaCl, 5%(v/v) glycerol, 1 mM TCEP, 20 μg $mL^{-1}$ DNase I, 5 mM $MgCl_2$, 1 mM PMSF, 10 mM imidazole, 50 mM Tris-HCl pH 8.0, and lysed with a cell disruptor. After clarification, the supernatant was stirred with 2 mL Ni-NTA resin at 4 °C for 2 h. The resin was washed with 50 mL buffer containing 150 mM NaCl, 5%(v/v) glycerol, 0.2 mM TCEP, 10 mM imidazole, 50 mM Tris-HCl pH 8.0. The protein was eluted with 250 mM imidazole, 150 mM NaCl, 5%(v/v) glycerol, 0.2 mM TCEP, 10 mM Tris-HCl pH 8.0, aliquoted, flash-frozen with liquid nitrogen and stored at −80 °C.

**Protein purification—sybodies**. The expression of sybodies in flasks was carried out by scaling-up the small-scale protocols. Cells were lysed by osmotic shock as follows. Biomass from 1 L of culture was resuspended in 20 mL TES buffer (0.5 M sucrose, 0.5 mM EDTA, and 0.2 M Tris-HCl pH 8.0) for dehydration at 4 °C for 0.5 h, followed by rehydration by dilution with 40 mL of ice-cold MilliQ $H_2O$ at 4 °C for 1 h. To collect the periplasmic extracts, the suspension was centrifuged at $25,000 \times g$ at 4 °C for 30 min and pellets were discarded. The supernatant was adjusted to have 150 mM of NaCl, 2 mM of $MgCl_2$, and 20 mM of imidazole. Ni-NTA resin (2 mL), pre-equilibrated with 20 mM imidazole (all buffer contained 150 mM NaCl and 20 mM HEPES pH 7.5), was added into the supernatant for batch binding with stirring at 4 °C for 1.5 h. The resin was packed into a gravity column and washed with 20 CV of 30 mM imidazole. The sybody was eluted using 300 mM imidazole.

**Expression of fluorescently tagged membrane proteins in *E. coli***. The expression of *aa*PlsY with different fluorescent protein tags was carried out as below[65]. BL21 (DE3) cells carrying p3EG-*aa*PlsY or pETSG-*aa*PlsY was cultured in Luria-Bertani (LB) medium (0.5%(w/v) yeast extract, 1%(w/v) peptone, 1%(w/v) NaCl) plus 50 mg $L^{-1}$ kanamycin in a 37 °C shaking incubator for overnight. This starter culture was 1:100 seeded into fresh LB-kanamycin medium for growth at 37 °C. When $OD_{600}$ reached 0.6, IPTG was added to a final concentration of 50 μM, and the temperature was dropped to 20 °C. After 16 h of induction, cells were harvested by centrifugation at $4000 \times g$ for 10 min at 4 °C, flash-frozen in liquid nitrogen, and stored at −80 °C until use.

msSI was expressed as for aaPlsY except that the strain C41 (DE3) was used instead of BL21 (DE3), and the induction was carried out using 0.1 mM IPTG at 25 °C. Note that msSI did not display isomerase activity in an established phenotype-rescuing assay but we still refer it as an SI.

**Expression of fluorescently tagged membrane proteins in S. cerevisiae.** S. cerevisiae BCY123 cells (MATα pep4::HIS3 prb::LEU2 bar1:HISG lys2::GAL1/10-GAL4 can1 ade2 ura3 leu2–3 112 trp1, lab collection) carrying p3YG-hSI or pYTSG-hSI was grown on SC-ura agar (76 mg L⁻¹ of Ala, Arg, Asn, Asp, Cys, Glu, Gln, Gly, His, Ile, Lys, Met, Phe, Pro, Ser, Thr, Trp, Tyr, and Val, 380 mg L⁻¹ of Leu, 18 mg L⁻¹ adenine, 8 mg L⁻¹ of p-aminobenzoic acid, 76 mg L⁻¹ myo-inositol, and 0.67%(w/v) YNB) supplemented with 2%(w/v) glucose. The authenticity of the yeast strain was checked by its auxotrophic phenotype. A single colony was inoculated into 2 mL of liquid SC-ura plus 2%(w/v) glucose in a 12-mL tube for culturing at 30 °C with shaking at 250 rpm. After 20 h, 0.3 mL of cells were diluted into 10 mL SC-ura supplemented with 0.1%(w/v) glucose to a final $OD_{600}$ of 0.12–0.15. Cells were grown to a density with $OD_{600}$ of 0.55–0.60 (~7–8 h), at which point the protein expression was induced at 20 °C at 250 rpm by the addition of 1.1 mL SC-ura supplemented with 20%(w/v) galactose. Cells were induced for 20 h before harvesting by centrifugation at 4000 × g for 10 min at 23 °C, flash-frozen and stored at −80 °C until use. The expression of PIS in S. cerevisiae followed the same procedures as hSI.

**Expression of fluorescently tagged membrane proteins in the baculovirus system (insect cells).** Genes of interest were cloned into p3FG or pFTSG, which were modified based on pFastBac (Invitrogen). Baculovirus was generated following the standard Bac-to-Bac protocol. For expression, Spodoptera frugiperda Sf9 suspension cells (lab collections, not checked for mycoplasma contamination) at 4 million per mL were infected with 1%(v/v) virus. After 48 h of incubation at 27 °C, cells from 10 mL of culture were harvested by centrifugation at 300 × g for 5 min at 4 °C. Cell mass was washed with 1 mL of PBS buffer (136 mM NaCl, 2.6 mM KCl, 2 mM $KH_2PO_4$, 8 mM $Na_2HPO_4$ pH 7.4), flash-frozen in liquid nitrogen, and stored at −80 °C until use.

**Expression of fluorescently tagged membrane proteins in mammalian Expi293F cells.** Suspension-adapted Expi293F (Cat. A14527, Thermo Fisher Scientific, Waltham, MA, USA) cells were used for transient expression. Cells were not checked for mycoplasma contamination. The procedure was the same for hSI and PIS. Forty micrograms of the plasmid pMG-hSI or pBTSG-hSI and 80 µg of linear 25-kDa polyethylenimine was incubated with 1 mL Union293 medium separately for 3 min at room temperature (19–22 °C). The two were then combined and incubated for 20 min before added into 20 mL of cells at a density of $2 × 10^6$ cells per mL. Cells were cultured in a 250-mL conical flask in a shaker at 37 °C with 5% (v/v) $CO_2$ and 130 rpm shaking. After 48 h, cells were collected by centrifugation at 300 × g for 5 min at 4 °C. Cell pellets were washed with 10 mL of PBS buffer (136 mM NaCl, 2.6 mM KCl, 2 mM $KH_2PO_4$, 8 mM $Na_2HPO_4$ pH 7.4), flash-frozen in liquid nitrogen, and stored at −80 °C until use.

**Membrane isolation and solubilization (for fluorescent protein-based FSEC-TS assay).** Membranes were isolated as follows. For E. coli, cells from 10 mL culture were resuspended in 1 mL of Buffer A (150 mM NaCl, 10%(v/v) glycerol, 0.1 mM EDTA, 0.2 mM TCEP, 0.2 × protease inhibitor cocktail, 50 mM Tris-HCl pH 8.0) for sonication using an ultrasonic homogenizer (Cat. SCIENTZ-2400F, Scientz, Ningbo, China) and a 2-mm probe at 30% power for 10 min in ice-water with on/off cycles of 3/5 s. For S. cerevisiae, cells from 10 mL of culture were resuspended in 1 mL of Buffer A, incubated with 0.4–0.5 g of 0.5-mm precooled glass beads (Cat. 150005 G, Aoran Technology, Shanghai, China), and mechanically broken using a Precellys Evolution homogenizer (Cat. P000062-PEVO0-A, Bertin, Germany) at 8000 rpm for 30 s for five times. Between each cycle, the cells were cooled down on ice for at least 3 min. Insect cell and mammal cells were lysed with solubilizing detergents. Membranes were isolated from the cell lysate by centrifugation at 55,000 rpm (~150,000 × g) using a Beckman TLA-55 rotor for 1 h at 4 °C.

aaPlsY was solubilized in buffer containing 1%(w/v) dodecylmaltoside (DDM), 150 mM NaCl, 10%(v/v) glycerol, 0.1 mM EDTA, 0.2 mM TCEP, and 50 mM Tris-HCl pH 8.0. msSI was solubilized in buffer containing 1%(w/v) DDM, 150 mM NaCl, 0.1 × protease cocktail, and 50 mM Tris-HCl pH 8.0. hSI was solubilized in the same buffer as aaPlsY. PIS was solubilized in buffer containing 0.9%(v/v) Triton X-100, 5%(v/v) glycerol, 0.5 mM DTT, 10 µM PMSF, 30 mM $MgCl_2$, 1 mM $MnSO_4$, and 20 mM Tris-HCl pH 8.5. ttSI and conSI were solubilized in 1%(w/v) DDM, 0.2%(w/v) CHS, 500 mM NaCl, 1 mM PMSF, 1 mM TCEP, 0.1 × protease inhibitor cocktail, 2 mM imidazole, 50 mM Tris-HCl pH 8.0.

**Stability assay—free TGP and GFPs.** Purified fluorescent proteins were heated at various temperatures and centrifuged at 21,000 × g for 30 min at 4 °C. The supernatant was either taken directly for fluorescence counting or FSEC analysis. For FSEC, the running buffer contained 0.03%(w/v) DDM, 150 mM NaCl, and 50 mM Tris-HCl pH 8.0. The fractional counts or peak intensities were plotted

against temperatures. Data from a single experiment were fitted with a sigmoidal dose-response function. The $T_m$ values reported are from the fitting.

**Stability assay—TGP- or GFP-based FSEC-TS for membrane proteins.** For FSEC-TS assay, 0.1 mL of solubilized fraction were heated at various temperatures on a PCR cycler for 20 min. The samples were cooled on ice for 10 min, and centrifuged at 21,000 × g for 30 min at 4 °C to remove precipitates before FSEC analysis. Depending on the expression yield, 1–5 µL of samples were loaded onto a Sepax Zenix-C SEC-300 column (Cat. 233300–4630, Sepax Technologies, Newark, DE, USA) connected to a Shimadzu HPLC machine equipped with a fluorescence detector (RF-20A, Shimadzu) with excitation/emission wavelength of 470/512 nm. FSEC running buffers were the following. aaPlsY and msSI, 0.03%(w/v) DDM, 0.2 mM TCEP, 150 mM NaCl, and 50 mM Tris-HCl pH 8.0. hSI, 0.03%(w/v) DDM, 0.1 mM EDTA, 0.2 mM TCEP, 150 mM NaCl, and 50 mM Tris-HCl pH 8.0, PIS: 0.1%(v/v) Triton X-100, 5%(v/v) glycerol, 0.5 mM DTT, 20 mM $MgCl_2$, 200 mM NaCl, and 20 mM Tris-HCl pH 8.5; ttSI and conSI, 0.02%(w/v) DDM, 0.004%(w/v), 150 mM NaCl, 0.1 mM EDTA, 0.25 mM TCEP, 0.1 × protease inhibitor cocktail, 20 mM HEPES pH 7.5; spPlsY, 0.03% LMNG, 150 mM NaCl, 50 mM Tris-HCl pH 8.5.

**Stability assay—tryptophan-based FSEC-TS for ttSI and hSI.** ttSI was purified for tryptophan-based FSEC-TS assay as follows. Membranes from 4 L of S. cerevisiae cells expressing ttSI-TGP[64] were solubilized in buffer containing 1%(w/v) DDM, 0.2%(w/v) CHS, 500 mM NaCl, 1 mM PMSF, 1 mM TCEP, 0.1 × protease inhibitor cocktail, 2 mM imidazole, 50 mM Tris-HCl pH 8.0 at 4 °C for 2 h with mild stirring. The solubilized material containing ttSI-TGP was collected as the supernatant of ultracentrifugation at 150,000 × g for 1 h at 4 °C. TALON beads (4 mL) were added to the supernatant for batch binding at 4 °C for 2.5 h. The beads were packed into a gravity column, and washed with 15 CV of buffer with 10 mM imidazole in buffer 0.03%(w/v) DDM, 0.006%(w/v) CHS, 1 M NaCl, 4%(w/v) glycerol, 0.05 mM EDTA, 0.2 mM TCEP, 0.1 × protease inhibitor cocktail, 20 mM HEPES pH 7.5. TGP-free ttSI was then released from the column by His-tag-free 3C protease digestion, concentrated using a 100-kDa cutoff filtration membrane, heated at various temperatures, and used for FSEC-TS assay with the same setup as for GFP fusion protein except that the excitation and emission wavelength pair was changed to 280/350 nm. The stability assay for hSI by tryptophan-FSEC was carried out exactly the same way as for ttSI except that the solubilization and purification steps were performed in the absence of CHS.

**Stability assay—$T_m$ measurement of spPlsY using its enzymatic activity.** To obtain $T_m$ using enzymatic assay, spPlsY was purified as follows. E. coli BL21 (DE3) cells carrying the plasmid p3EC-spPlsY[65] (PlsY from Staphycoccus pneumonia) were induced with 50 µM IPTG for 16 h in M9 medium[65]. Membranes were solubilized using 1%(w/v) DDM in Buffer SP (10% (v/v) glycerol, 0.2 mM TCEP, 0.15 M NaCl and 50 mM Tris-HCl pH 8.0). The supernatant, clarified by centrifugation of the solubilized sample at 48,000 × g at 4 °C for 1 h, was incubated with Ni-NTA beads for batch binding at 4 °C for 1 h with mild stirring. The beads were washed with 20 mM imidazole in 0.03%(w/v) LMNG for detergent exchange. spPlsY was eluted using 250 mM imidazole and 0.03%(w/v) LMNG in Buffer SP.

To determine the $T_m$, spPlsY was heated for 20 min at various temperatures. The Pi-releasing acyltransferase activity was then measured using a coupled assay with a fluorescently labeled phosphate-binding protein (PBP)[83,84]. The assay mix contained 30 mM glycerol 3-phosphate (substrate 1), 30 ng mL⁻¹ of spPlsY, 20 µM of palmitoyl phosphate (substrate 2), and 5 µM MDCC-PBP in the buffer containing 0.03%(w/v) LMNG, 0.15 M NaCl, 50 mM Tris-HCl pH 8.0. The increase of fluorescence (excitation and emission wavelength of 425/466 nm) as the readout of spPlsY activity was monitored at 30 °C in a plate reader for 1 h at 30 s intervals. The percentage activity to the unheated sample was plotted against temperatures for curve fitting.

**Yield calculation—fluorescence counting.** Cells from 1 mL of culture were collected in a 1.5-mL Eppendorf tube by centrifugation at 12,000 × g for 1 min at 4 °C. Cell pellets were resuspended in 0.2 mL of buffer (150 mM NaCl, 50 mM Tris-HCl pH 8.0) for fluorescence measurement in a black 96-well plate in a SpectraMax M2e plate reader (Molecular Devices, San Jose, CA, USA) with excitation/emission wavelength pair of 488/512 nm. The fluorescence counts corrected by dilution factors were used to calculate expression yield based on a calibrated linear quantity-intensity relationship of fluorescent protein standards (Fig. 1a). Three independent experiments were performed using different colonies or transfection of cells with different passage numbers.

**Yield calculation—in-gel fluorescence.** Membranes were solubilized in SDS-loading buffer for electrophoresis. SDS-PAGE gels were imaged directly using a ChemiDoc MP (Bio-Rad) with the GFP-channel setting. Fluorescence markers prepared in-house[14] were included alongside of membrane protein fusion samples. Band intensities were integrated using the software Image Lab (Bio-Rad). The relative yield was calculated using integrated intensities applied with the correction factors based on the quantity-intensity relationship for each fluorescent protein.

Three independent experiments were performed using different colonies or transfection of cells with different passage numbers.

**Yield calculation—Yield$_{FSEC}$**. To calculate Yield$_{FSEC}$, the relative fluorescence intensity of the same amount of the three fluorescent proteins were calibrated using analytic FSEC with signals at the excitation/emission wavelength of 470/512 nm. The relative intensity was at 1: 0.91: 0.33 (TGP: ecGFP: scGFP). This correction factor was applied to the peak intensities before calculating the relative Yield$_{FSEC}$. For example, a ratio of fluorescence intensity at 2: 1 translates to a Yield$_{FSEC}$ ratio of 0.66: 1 (TGP: scGFP). Three independent experiments were performed using different colonies or transfection of cells with different passage numbers.

**Sybody selection—ribosome display and phage display**. To facilitate sybody selection, TGP was engineered to contain an Avi-tag (GLNDIFEAQKIEWHE), followed sequentially by the 3C protease site (LEVLFQGP), and an octa-histidine tag at the C-terminus. This form (TGP$_{Avi}$) was purified the same way as described above. For biotinylation, 4.5 mg mL$^{-1}$ of TGP$_{Avi}$ was incubated with 0.13 mg mL$^{-1}$ of His-tagged BirA, 0.72 mg mL$^{-1}$ His-tagged 3C protease, 5 mM ATP, 10 mM magnesium acetate, 0.23 mM biotin in 500 µL volume and incubated at 4 °C for 11 h. The solution was incubated with 0.5 mL pre-equilibrated Ni-NTA resin and rotated at 4 °C for 1.5 h to remove His-tagged 3C protease and BirA. The flow-through fraction containing biotinylated TGP$_{Avi}$ (free of His-tag) was loaded onto a Superdex 200 increase 10/300 GL column pre-equilibrated with buffer containing 150 mM NaCl, 20 mM Tris-HCl pH 8.0 to remove ATP and biotin. Fractions with A$_{493}$ were pooled, adjusted to 0.6 mg mL$^{-1}$, aliquoted, flash-frozen with liquid nitrogen, and stored at −80 °C.

In vitro translation of the 'concave' library[59] was performed as follows. A mix (PURE*frex* 2.1 kit, Cat. PF213-0.25-EX, Genefrontier, Chiba, Japan) containing 1.8 µL nuclease-free water, 4 µL of solution I and 0.5 µL solution II and 1 µL solution III, 0.5 µL 10 mM cysteine, 0.5 µL 80 mM reduced and 0.5 µL 60 mM oxidized glutathione, and 0.5 µL 1.875 mg mL$^{-1}$ of the disulfide bond isomerase DsbC (DS supplement, Cat. PF005-0.5-EX, Genefrontier) was incubated at 37 °C for 5 min in a PCR cycler. mRNA library (0.7 µL, corresponding to 1.6 × 10$^{12}$ mRNA molecules) was added to the pre-warmed mix for translation at 37 °C for 30 min. The products were diluted with 100 µL ice-cold panning solution containing 150 mM NaCl, 50 mM magnesium acetate, 0.05%(w/v) BSA, 0.1%(w/v) Tween 20, 0.5%(w/v) heparin, 1 µL RNaseIn (RNase inhibitor), and 50 mM Tris-acetate pH 7.4. The solution was cleared by centrifugation at 20,000 × g for 5 min at 4 °C. Biotinylated TGP$_{Avi}$ was added to the supernatant for the panning on ice for 20 min. Beads coated with streptavidin (Dynabeads Myone Streptavidin T1) was added to pull down the complex consisting of nascent sybodies, the ribosome with the mRNA encoding the binders, and biotinylated TGP$_{Avi}$. Enriched mRNAs were purified and reverse-transcripted with the primer 5′-CTTCAGTTGCCGC TTTCTTTCTTG-3′. The resulting cDNA library was purified and amplified using primer pairs 5′-ATATGCTCTTCTAGTCAGGTTCAGCTGGTTGAGAGCG-3′ and 5′-TATAGCTCTTCATGCGGCTCACAGTCACTTGGGTACC-3′. The product was gel-purified, digested with *Bsp*QI, and ligated into the vector pDX_init digested with *Bsp*QI. The ligation product was transformed into SS320 competent cells by electroporation to generate a library for phage display.

The first round of phage display was performed in a 96-well plate coated with 60 nM neutravidin (Cat. 31000, Thermo Fisher Scientific, Waltham, MA, USA). Phage particles were incubated with 50 nM biotinylated TGP$_{Avi}$, washed, and released from the beads by tryptic digestion with 0.25 mg mL$^{-1}$ trypsin in buffer 150 mM NaCl, 20 mM Tris-HCl pH 7.4. The enriched phage particles were amplified and screened the second and third round phage display using 12 µL MyOne Streptavidin T1 Dynabeads as the immobilizing matrix. TGP$_{Avi}$ concentration was added to 50 nM and 5 nM for the second and third panning rounds. All the panning procedures were carried out in the buffer contained 0.1% (w/v) Tween 20, 150 mM NaCl, 50 mM Tris-HCl pH 7.4. After selection, the phagemid containing enriched sybodies were subcloned into pSb_init vector using fragment-exchange (FX) cloning and transformed into *E. coli* MC1061 for expression and purification.

**Sybody selection—small-scale pull-down assay to identify TGP binders**. To screen TGP binders at a single-colony level, sybodies expressed in the periplasmic space were extracted first as follows. MC1061 single colonies carrying pSb-init-sybody plasmids were inoculated into 1 mL of terrific broth (TB) supplemented with 25 µg mL$^{-1}$ chloramphenicol in a 2-mL 96-well plate. Cells were grown for 5 h at 37 °C in a shaking incubator at 300 rpm before seeded into a new 96-well plate with 1:20 dilution. Two hours later, the temperature was shifted to 22 °C and the cells were cultured for another 1.5 h before induced with 0.02%(w/v) arabinose for 16 h. During the induction, the plate was sealed with a gas-permeable seal (Cat. 740675, Macherey-Nagel, Düren, Germany). Cells were harvested in the plate by centrifugation at 3220 × g for 30 min at RT. Cell pellets were resuspended in 100 µL of buffer containing 0.5 µg mL$^{-1}$ lysozyme, 20%(w/v) sucrose, 0.5 mM EDTA, and 50 mM Tris-HCl pH 8.0 and incubated for 30 min with shaking. The suspension was diluted by adding 400 µL of buffer containing 150 mM NaCl, 1 mM MgCl$_2$, and 50 mM Tris-HCl pH 7.4, and clarified by centrifugation at 3220 × g for 30 min at 4 °C.

Small-scale pull-down assays were carried out in a 1.1 mL-deep-well plate to pre-screen TGP binders. To 200 µL of periplasmic extraction, 5 µL Ni-NTA resin was added for binding of His-tagged sybodies. After shaking at 600 rpm for 30 min, the resin was settled down by gravity. The supernatant was removed carefully using a multi-channel pipette. The resin was then washed twice with 0.5 mL of wash buffer (20 mM imidazole, 150 mM NaCl, 50 mM Tris-HCl pH 8.0). To the resin, 200 µL of 30 µg mL$^{-1}$ of purified strep-tagged TGP (from N- to C-terminus, MGSKLSGREFLEGT-LEVLFQGP-(GGS)$_2$-TGP-(SGGG)$_2$-WSHPQFEK) was added. Batch binding and washes were repeated as above. TGP-sybody complexes were eluted from the resin with 200 µL of elution buffer (500 mM imidazole, 150 mM NaCl, 50 mM Tris-HCl pH 8.0). TGP fluorescence was measured with 100 µL of elution in a black 96-well plate under 'top-read' mode to avoid interference from beads. As a positive control, a GFP nanobody[14] fused to GST was immobilized to glutathione resin to pull down His-tagged GFP in the same setup. A sybody against the maltose-binding protein[59] was used as a negative control.

**Sybody selection—FSEC assay to screen TGP binders**. Periplasmic extracts described above were mixed with 1 µM of TGP. One microliter of the mixture was loaded onto a Sepax Zenix-C SEC-300 column (Cat. 233300–4630, Sepax Technologies, Newark, DE, USA) for analytical FSEC in the running buffer containing 150 mM NaCl and 50 mM Tris-HCl pH 8.0. A sybody against the maltose-binding protein[59] was used as a negative control.

**Biolayer interferometry assay**. To measure kinetics for TGP-sybody binding using biolayer interferometry assay with an Octet system, biotinylated TGP$_{avi}$ was immobilized onto a streptavidin SA sensor (ForteBio, Cat 18-5019) by soaking the sensor in TGP$_{avi}$ (2 µg mL$^{-1}$) solution in a buffer containing 0.005%(v/v) Tween 20, 150 mM NaCl, and 20 mM HEPES pH 7.5. After a balance phase in sybody-free buffer, SA sensors were incubated in various sybody concentrations at 7.4, 22, 66, and 200 nM in the same buffer (Association phase). For Sb92, only the last three concentrations were used. For the dissociation phase, the sensors were put in sybody-free buffer again. The data was fitted using a 1:1 stoichiometry with the built-in software Data Analysis 10.0.

**Purification of *ms*SI-TGP—by conjugated sybody 44**. Sb44 resin was prepared by incubating Sb44 (4 mg) with 1 mL of swelled CNBr-activated Sepharose 4B beads (Cat. 17043001, GE Healthcare) pre-equilibrated with buffer contained 0.5 M NaCl, and 0.1 M NaHCO$_3$ pH 8.3 at 4 °C for 9 h. After amine coupling, the beads were blocked by 10 mL of 0.1 M Tris-HCl pH 8.0. The beads were then washed extensively with 150 mM NaCl, 20 mM HEPES pH 7.5 before use.

Cells expressing *ms*SI-TGP were disrupted using a Constant cell disruptor. The cell lysate was clarified by centrifugation at 20,000 × g at 4 °C for 30 min. Membranes were sedimented by ultracentrifugation at 150,000 × g at 4 °C for 1 h, and resuspended in 16 mL of solubilized buffer (1%(w/v) DDM, 150 mM NaCl, 10%(v/v) glycerol, 1 mg mL$^{-1}$ iodoacetamide, 0.2 mM TCEP, 0.1 × protease inhibitor cocktail, 50 mM Tris-HCl pH 8.0). After 2 h of mild agitation at 4 °C for solubilization, unsolubilized pellets were discarded by centrifugation at 48,000 × g for 1 h at 4 °C. The supernatant was loaded onto 0.6 mL of CNBr-activated Sepharose 4B resin on a gravity column. The resin was then washed with 20 column volume (CV) of Buffer MS (0.03%(w/v) DDM, 0.2 mM TCEP, 5%(v/v) glycerol, 150 mM NaCl, 50 mM Tris-HCl pH 8.0). *ms*SI was released from the column by resuspending the beads in the buffer above supplemented with 1:40 (mol:mol) of 3C protease. The digestion was carried out at 4 °C overnight. The flow-through fraction containing tag-free *ms*SI and the His-tagged 3C protease was incubated with Ni-NTA to remove the protease.

**Purification of *ms*SI—by immobilized metal affinity chromatography**. *ms*SI was expressed in *E. coli* as an N-terminally His-tagged protein and solubilized as above. To clarified solubilized fraction, 4 mL TALON resin and 5 mM imidazole was added for batch binding for 2 h at 4 °C. The resin was packed into a gravity column and washed with 15 CV of 15 mM imidazole in Buffer MS. His-tagged *ms*SI was eluted using 200 mM imidazole in Buffer MS. *ms*SI was freed of His-tag by 3C protease digestion at 4 °C for 16 h. The mix was incubated with Ni-NTA resin to remove the His-tagged 3C protease and the flow-through fraction was collected.

**Crystallization of the TGP-Sb44 complex**. The TGP construct for crystallization contained residues 1–218 of TGP with a C-terminal octa-histidine tag spaced with a (SGGG)$_2$ linker. TGP and sybody were mixed at a molar ratio of 1:1.2, and the complex was purified using SEC in 150 mM NaCl, 20 mM HEPES pH 7.5. Fractions of the complex were pooled, concentrated to 20 mg mL$^{-1}$ using 10-kDa cutoff filtration membranes (Cat. UFC501096, Merck Millipore, Burlington, MA, USA). Sitting drop crystallization was performed by first pipetting 70 µL of precipitant solution into each well as reservoir, followed by depositing 150 nL of the pre-cipitant solution on top of 150 nL of protein with a Crystal Gryphon LCP robot (Art Robbins Instruments). Crystal plates were incubated at 20 °C in a Rock Imager R1000. Crystals were obtained in condition containing 0.2 M ammonium acetate, 25%(w/v) polyethylene glycol 3,350, 0.1 M Bis-Tris pH 6.5.

**Data collection and structure determination**. Crystals were transferred sequentially from sitting drop wells to 2-μL drops (on a glass slide) of mother liquor supplemented with 2.5, 5, and 7.5%(v/v) glycerol using a MiTeGen loop (Cat. M5-L18SP series, Ithaca, NY, USA). Crystals were allowed to equilibrate with the glycerol gradients for 5–10 s before sequential transfer. Cryo-cooling was performed by the rapid plunge of cryo-protected crystals into liquid nitrogen. Crystals were screened by X-ray diffraction at beamlines 18U1 and 19U1 at Shanghai Synchrotron Radiation Facility. Diffraction data were collected with a $50 \times 50$ μm beam on a Pilatus detector at a distance of 300 mm, with oscillation of 0.5° and a wavelength of 0.97930 Å. Data were integrated using XDS[85], scaled and merged using Aimless[86]. The structure was solved by molecular replacement using Phaser[87] with a TGP monomer from PDB 4TZA[58] and the sybody from PDB 6QV1[78] as the searching templates. The model was built with $2F_o$-$F_c$ maps in Coot[88], and refined using Phenix[89]. Structure was visualized in PyMol[90].

## Statistics and reproducibility

All experiments for the measurement of expression levels were replicated three times independently. The FSEC-TS data in Fig. 2e were from four independent experiments. Independent experiments refers to experiments starting with different bacterial and yeast colonies, or cells with different passage numbers for insect cells and mammalian cells. Statistical analyses were performed using GraphPad Prism 8. Error bars in all figures refer to standard deviation.

**Reporting summary**. Further information on research design is available in the Nature Research Reporting Summary linked to this article.

## Data availability

Atomic coordinates and structure factors for the reported TGP-Sb44 structure are deposited in the Protein Data Bank (PDB) under accession codes of 6LZ2. Relevant plasmids and sequences have been deposited in Addgene (www.addgene.org) with the following IDs: pETSG, 159418; pYTSG, 159419; pFTSG, 162389; pBTSG, 159420; pSB_init_Sb44, 159421; pSB_init_Sb66, 159422; pSB_init_Sb68, 159423; pSB_init_Sb92, 159424. Source data for Figs. 1a,c,d,e, Fig. 2, Figs. 3a,b,e–h, Figs. 4a,b,e–h, Figs. 5a,c,d,e,g, h, Figs. 6a,c,d,e,g,h, and Figs. 7c–f are available through Supplementary Data 1. There are no restrictions on data availability.

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

## Acknowledgements

We thank Professors Jinqiu Zhou, Ying Gao, and Yongning He at the authors' institute for providing relevant plasmids, yeast strains, and cell lines. We thank Dr. Juan Bao for assistance in molecular cloning, the staff members of the Large-scale Protein Preparation System at the National Facility for Protein Science in Shanghai (NFPS), Zhangjiang Lab for equipment maintenance and management, and staff scientists at BL18U1 and BL19U1 beamlines of NFPS at Shanghai Synchrotron Radiation Facility, for assistance during data collection. This research has been supported by the Strategic Priority Research Program of the Chinese Academy of Sciences (CAS) (XDB37020204), National Natural Science Foundation of China (31870726), Key Program of CAS Frontier Science (QYZDB-SSWSMC037), CAS Facility-based Open Research Program, and Shanghai Science and Technology Commission general program (20ZR1466700).

## Author contributions

H.C. and H.Y. performed protein expression, purification, characterization, and crystallization. H.C. and T.L carried out sybody screening under the supervision of C.A.J.H. and M.A.S. Y.L. performed crystallization and assisted data collection. Y.T. performed experiments for *sp*PlsY. D.L. conceived the research, collected X-ray diffraction data, solved the structure, and wrote the manuscript. M.A.S. edited the manuscript.

## Competing interests

The authors declare no competing interests.
