## [Peer Review File · Communications Biology]

Reviewers' Comments:

Reviewer #1:

Remarks to the Author:

The isolation of membrane proteins for functional and structural investigation is an empirical process. The GFP-based tagging of membrane proteins has become a common approach to facilitate these optimisation processes. The paper by Li and co-workers shows that the GFP most commonly used from jellyfish can be replaced by a version from coral (TGP). The advantage with the coral version (TGP) is that it is more thermostable than standard GFP.

Overall, this is a clearly written paper and the results are well presented. Methods take a long time to develop and the authors have tested the TGP version thoroughly. However, since standard GFP is already very stable it seems a lot of effort for a small amount of gain. For example, the authors mention that you cannot measuring melting temperatures for membrane protein-GFP fusions by FSEC when the T_m is more than 70C, but with TGP you can measure up to 90C. Since most membrane proteins have a T_m around 40 to 50C its only a useful for a handful of membrane proteins that happen to have very high melting temperatures. In those small cases since the protein is already very happy ($> 70C T_m$) then you are not going to be using FSEC to help optimise stability as its already very good. The main reason for wanting to be able to measure T_m for proteins higher than 70C if is you want to use the fluorescent tag to monitor ligand binding. However, experiments to show that the TGP fusion was also compatable for measuring ligand binding by FSEC-TS was not shown.

Minor comments:

1. The slybody is a very useful addition here. Does the slybody efficiently bind to eGFP as well?
2. It was not exactly clear to me what version scGFP was? I looked up the NCBI accession number given in the paper and it was referred to as "green fluorescent protein (mitochondrion) [synthetic construct]". At least the version used in *S. cerevisiae* paper mentioned is a yellow-shifted variant that has a different excitation wavelength than GFP and would explain the differences in in-gel fluorescent intensities.
3. I would be a bit careful about the statement that this GFP version improves mammalian cell expression. We were excited about the previous publication showing that of different fluorescent protein fusions mVenus could increase yeilds (Rana, M. S., Wang, X. & Banerjee, A. An improved strategy for fluorescent tagging of membrane proteins for overexpression and purification in mammalian cells. *Biochemistry* 57, 6741-6751 (2018). However, after recloning several of our targets we didnt see any improvement when mVenus was used as a membrane protein fusion in HEK293 cells (unpublished). In the experiments shown here you have transfected HEK cells with the same concentration of plasmid? Did you also check to make sure that the difference in yields is not simply the result that there is a different DNA concentration optimum between the pMG-hSI or pBTSG-hSI plasmids?

Reviewer #2:

Remarks to the Author:

This manuscript describes the application of TGP-fusion technology for membrane protein expression and development of nanobodies for TGP. The manuscript is well-written and the standard of the work is quite high. I believe that the methods presented here would be very useful for scientists in the field of membrane protein structural biology, biochemistry and biophysics. Overall, I high recommend this manuscript for publication if the following concerns are properly addressed.

Major concerns

My major concern is whether TGP-fusion may yield more false positive hits in FSEC screening. As

the authors would know, GFP-tagged membrane proteins could sometimes precipitate after the removal of GFP by site-specific protease during purification. It may be often considered that GFP is a stable protein so that GFP fusion may sometimes "stabilize" or "protect" fused membrane proteins, which are unstable alone.

I fully understand the benefits of TGP use in FSEC. However, since TGP is even more stable than GFP, I have a concern that TGP may cause similar (or potentially even more) false positive hits. In FSEC screening.

To address this concern, the authors should measure and show the T_m values of purified hSI and PIS with and without TGP-tag by TRP-FSEC, since hSI and PIS showed much higher expression level with TGP. Investigation of whether and how hSI and PIS are stable in TGP-tagged and non-tagged form would be enough to address my above-mentioned concern.

It should be noted that I would still strongly support the publication of the manuscript even if non-TGP tagged hSI and PIS are much less stable than those with TGP tags. In this case, the authors can simply mention the false positive hit issue from TGP-fusion proteins as a possible pitfall of TGP-fusion technology in the discussion section. As long as the authors measure and show the T_m values of purified hSI and PIS with and without TGP-tag by TRP-FSEC for comparison, I would support the publication of the manuscript.

Minor concerns

1. Page 1: "Current strategies for membrane protein (MP) structure studies require functional isolation in large quantities but their hydrophobicity brings great challenges in almost every step such as expression, purification, and crystallization".

The authors should mention Cryo-EM techniques as a method for structure determination by citing a couple of articles, as Cryo-EM is becoming more common these days.

2. If there is no patent or license issue, I highly recommend the authors to deposit their vectors (pETSG, pYTSG, pBTSG, Sb44, Sb66 and Sb68) to Addgene so that their method can be more widely-used. I believe it would highly benefit the community. After deposition, the authors should show the Addgene IDs in the revised manuscript.

Reviewer #3:

Remarks to the Author:

In this manuscript by Cai Yao and Li describes the use of thermostable GFP (TGP) as a fusion protein for membrane proteins (MP). Fusion of a fluorescent protein is useful in structural studies where scanning multiple variables such as species choice, construct design, extraction detergent, and buffer is required in order to boost protein expression levels and enable crystallization. Since stable proteins tend to crystallize better, it is advantageous to identify mutations that increase the thermostability of the MP. When testing the stability of each construct and each condition it is beneficial to have a thermostable fluorescent fusion protein that will report on the state of the MP rather than undergo unfolding itself. In this study the authors applied a new fluorescent protein, TGP, to the current system of fluorescence detection size exclusion chromatography thermostability assay. The authors tested TGP as a fusion protein in three expression systems, mammalian, yeast, and bacteria and showed that in addition to faithfully report on the T_m of the MP, the fusion also help increase expression levels as compared to the same MP fused to GFP. The authored also generated synthetic nanobodies (sybodies) against TGP to facilitate purification. The authors also solved a crystal structure of TGP in complex with one of these sybodies.

This study provides a method advance to allow selection of more thermostable membrane protein primarily for structural studies. There are some technical points to consider, but otherwise the study is well-executed and does not overstate its finding.

Major points:

1. The study was done in three expression systems, mammalian, yeast, and bacteria. While mammalian cell and bacteria are frequently used in structural biology, hardly any large-scale protein production is done in *Saccharomyces cerevisiae*. In order for this study to be truly useful for the structural biology community it would be better to choose a more widely used expression system for membrane proteins such as Sf9 insect cells. There are advantages to yeast, such as the ease of cloning, but even though generating baculovirus is more demanding it will better reflect how this method will be used in a lab.

2. The authors used Isothermal titration calorimetry (ITC) to measure the sybodies affinity to TGP. While ITC has its merits, the gold-standard technique for measuring affinity of antibodies and antibody-fragments is Surface Plasmon Resonance (SPR). In addition to K_d , K_{off} and K_{on} are also measured thus providing additional important kinetic information. This will help in selecting a superior sybody that shows both a long K_{off} and low K_d for TGP.

Minor points:

1. The authors show that fusion of TGP gives higher yield as compared to fusion to GFP. It would be useful to show yields of the unfused MP for comparison. Some fusion proteins actually interfere with proper folding of their partner and thus give lower yields.

2. In the comparison of the melting profile of spPlsY using both enzymatic and fluorescence it is mentioned that the enzymatic activity dropped gradually upon heating (which is typical) while the fluorescence increased before dropping (lines 125-132 and Fig.2E). This was done only once and so there are no error bars. This experiment should be repeated a few times to generate error bars and asserting whether this is just a one-time occurrence or is this an actual phenomenon. This will provide the audience of this paper with some knowledge when they try to implement this method.

3. The selection of sybodies generated three viable options. The authors chose Sb44 which, at $K_d=109$ nM has the worst affinity for TGP. If there were other considerations for this choice, such as poor yield or folding issues for the other sybodies, they should be explicitly stated.

4. In the paragraph between lines 407-412 the authors suggest testing whether the sybodies bind to the same epitope. The authors also suggest two very straight forward and simple experiments to test this hypothesis. I encourage the authors to pursue this, since as they state in the same paragraph, this can be a very useful piece of information.

5. In the Methods section in lines 458-465 the contents of the lysis buffer are not stated.

6. For clarity, I think it is best to change the title in line 554 (Stability assay-TGP-or GFPbased FSEC-TS) to "membrane isolation and protein purification" or something along these lines for the paragraphs between lines 555-573. The original title should be applied to the paragraph between lines 575-586.

7. Line 594 change "was" to "were"

8. Line 709 change "gravitation" to "gravity"

9. In lines 734-737 the authors describe preparation of affinity resin by coupling Sb44 to CNBr resin. To make this point clear to the reader the section should start with something like "Sb44 resin was prepared by ..."

10. In line 766 it is not clear that the 70 ul refers to the volume of the reservoir. The authors should change the phrasing to make this clear to the reader.

Reviewers' comments:

Reviewer #1 (Remarks to the Author):

The isolation of membrane proteins for functional and structural investigation is an empirical process. The GFP-based tagging of membrane proteins has become a common approach to facilitate these optimisation processes. The paper by Li and co-workers shows that the GFP most commonly used from jellyfish can be replaced by a version from coral (TGP). The advantage with the coral version (TGP) is that it is more thermostable than standard GFP.

Overall, this is a clearly written paper and the results are well presented. Methods take a long time to develop and the authors have tested the TGP version thoroughly. However, since standard GFP is already very stable it seems a lot of effort for a small amount of gain. For example, the authors mention that you cannot measuring melting temperatures for membrane protein-GFP fusions by FSEC when the T_m is more than 70C, but with TGP you can measure up to 90C. Since most membrane proteins have a T_m around 40 to 50C its only a useful for a handful of membrane proteins that happen to have very high melting temperatures. In those small cases since the protein is already very happy ($> 70C T_m$) then you are not going to be using FSEC to help optimise stability as its already very good. The main reason for wanting to be able to measure T_m for proteins higher than 70C if is you want to use the fluorescent tag to monitor ligand binding. However, experiments to show that the TGP fusion was also compatable for measuring ligand binding by FSEC-TS was not shown.

-- We thank the reviewer for the supportive comments as well as the constructive critics.

We agree with the reviewer that in most cases, GFP is sufficient for T_m measurement.

We have attempted to apply the FSEC- T_m method to screen ligands for some lipid enzymes including hSI. However we have been unable to find any ligands that could increase the apparent T_m noticeably. This is certainly an interesting aspect of the method to explore in the future.

Minor comments:

1. The sybody is a very useful addition here. Does the sybody efficiently bind to eGFP as well?

-- No, they do not bind eGFP at all. These nanobodies are very specific.

Before we started the sybody selection project, we constructed two chimeric TGPs by 'copying' the epitope loops from ecGFP to TGP, expecting that the ecGFP nanobodies (enhancer and minimizer) to recognize the chimeric TGPs. However, no binding were observed, indicating high specificity.

2. It was not exactly clear to me what version scGFP was? I looked up the NCBI accession number given in the paper and it was referred to as “green fluorescent protein (mitochondrion) [synthetic construct]”. At least the version used in *S. cerevisiae* paper mentioned is a yellow-shifted variant that has a different excitation wavelength than GFP and would explain the differences in in-gel fluorescent intensities.

We have changed the NCBI accession code to *ABI82039.1* because the description (yeast enhanced green fluorescent protein) is more relevant to the current study. We note that the sequence of *ABI82039.1* is the same as the previous code (*AAB51348.1*). For your information, the amino acid sequence is attached below.

The excitation wavelength of scGFP was indeed yellow-shifted (Fig. R1). To reflect this, we have added the following sentence to the revised manuscript:

In addition, the scGFP contains mutations²⁷ that cause yellow-shift of the excitation wavelength ($\lambda_{\max} = 502$ nm), which is partly responsible for its relatively low intensity in both counting and in-gel fluorescence analysis.

Fig. R1. Normalized absorption spectrum of scGFP (blue), ecGFP (orange), and TGP (grey). The λ_{\max} values are as indicated.

>scGFP

```
MSKGEELFTGVVPIVLVDGDVNGHKFSVSGEGEGDATYGKLTCLKFICTTGKLPVPWPTL
VTTFGYGVQCFARYPDHMKQHDFFKSAMPEGYVQERTIFFKDDGNYKTRAEVKFEEDT
LVNRIELKGIDFKEDGNILGHKLEYNYNSHNVYIMADKQKNGIKVNFKIRHNIEDGSVQL
ADHYQQNTPIGDGPVLLPDNHYLSTQSALS KDPNEKRDHMLLEFVTAAGITHGMDELY
K
```

3. I would be a bit careful about the statement that this GFP version improves mammalian cell expression. We were excited about the previous publication showing that of different fluorescent protein fusions mVenus could increase yields (Rana, M. S., Wang, X. & Banerjee, A. An improved strategy for fluorescent tagging of membrane proteins for overexpression and purification in mammalian cells. *Biochemistry* 57, 6741-6751 (2018). However, after recloning

several of our targets we didn't see any improvement when mVenus was used as a membrane protein fusion in HEK293 cells (unpublished). In the experiments shown here you have transfected HEK cells with the same concentration of plasmid? Did you also check to make sure that the difference in yields is not simply the result that there is a different DNA concentration optimum between the pMG-hSI or pBTSG-hSI plasmids?

-- We thank the reviewer to share his/her experiences with the mVenus. We agree that the effect on yield is vitally important for a method paper. Therefore, we carried out the comparison as briefly described below.

First, we determined the optimal plasmid concentrations (1, 2, 3, and 4 mg L⁻¹) for both constructs. The expression of hSI-scGFP increased at increasing concentrations of plasmid in the range of 1, 2, and 3 mg L⁻¹ and leveled off at 3 mg L⁻¹. By contrast, the expression level of hSI-TGP leveled off at 2 mg L⁻¹.

When assessed under their respective optimal conditions, hSI-TGP showed a ~10 folds expression level compared with hSI-scGFP.

In the previous manuscript, the expression level was compared at a plasmid concentration of 2 mg L⁻¹. Under this condition, the expression level of hSI-TGP was over 30 folds than hSI-scGFP. We note that this result was reproducible during the optimization experiment mentioned above.

With the new results, we have modified the original results as the following:

TGP showed remarkable improvement for hSI expression in the mammalian system. First, for maximal transient expression level, lower amount of plasmid was required for hSI-TGP (2 mg L⁻¹) than for hSI-scGFP (3 mg L⁻¹). When assessed under optimal conditions, the yield for hSI-TGP was over 10 folds compared with hSI-scGFP using the three aforementioned quantitative methods (**Fig. 6A-6D, Table 1**).

Accordingly, the original data in Fig. 5A-5D have been replaced with new data in Fig. 6A-6D in the revised manuscript.

Reviewer #2 (Remarks to the Author):

This manuscript describes the application of TGP-fusion technology for membrane protein expression and development of nanobodies for TGP. The manuscript is well-written and the standard of the work is quite high. I believe that the methods presented here would be very useful for scientists in the field of membrane protein structural biology, biochemistry and biophysics. Overall, I high recommend this manuscript for publication if the following concerns are properly addressed.

-- We thank the reviewer for the supportive comments to our manuscript.

Major concerns

1. My major concern is whether TGP-fusion may yield more false positive hits in FSEC screening. As the authors would know, GFP-tagged membrane proteins could sometimes precipitate after the removal of GFP by site-specific protease during purification. It may be often considered that GFP is a stable protein so that GFP fusion may sometimes “stabilize” or “protect” fused membrane proteins, which are unstable alone.

-- We agree with the reviewer on this concern, as we have indeed observed this during membrane protein purification with GFP/TGP tags.

We have modified our previous discussions to reflect this.

“As an indirect assay, how *apparent* T_m values would differ from *actual* T_m values (as determined by functional assays) is always a topic of general interest. It is widely accepted that such apparent T_m values should be treated with caution. Only when denaturation coincides with aggregation would the values be the same. Occasionally, membrane proteins, even for those with a high apparent \$T_m\$ in the FP-fused form, precipitate upon FP removal, suggesting protective effect of these stable tags on membrane proteins. Such effects, if not affecting membrane protein fold and activity, may be advantageous for cryo-EM studies because density from flexible tags could be averaged out during reconstruction so the data may be acquired using FP-tagged protein. For crystallization, it may require changing to FP-free constructs so that the POI did not experience compositional shock during later purification steps. Despite these limitations, FSEC-TS should be considered when absolute \$T_m\$ values are not necessarily required, for example, in the screening of stable homologs or mutants for structural study, or screening of lipids, ligands for purification, owing to its simplicity, general applicability, and high-throughput feature.”

2. I fully understand the benefits of TGP use in FSEC. However, since TGP is even more stable than GFP, I have a concern that TGP may cause similar (or potentially even more) false positive hits. In FSEC screening.

To address this concern, the authors should measure and show the T_m values of purified hSI and PIS with and without TGP-tag by TRP-FSEC, since hSI and PIS showed much higher expression

level with TGP. Investigation of whether and how hSI and PIS are stable in TGP-tagged and non-tagged form would be enough to address my above-mentioned concern.

It should be noted that I would still strongly support the publication of the manuscript even if non-TGP tagged hSI and PIS are much less stable than those with TGP tags. In this case, the authors can simply mention the false positive hit issue from TGP-fusion proteins as a possible pitfall of TGP-fusion technology in the discussion section. As long as the authors measure and show the T_m values of purified hSI and PIS with and without TGP-tag by TRP-FSEC for comparison, I would support the publication of the manuscript.

-- We thank the reviewer for the constructive comments.

We have performed the experiment suggested by the reviewer for both hSI and *tt*SI (Fig. 2D, below). The thermal denaturation profiles were virtually the same between TGP-FSEC and Trp-FSEC.

We did the same experiment for PIS. However, the TGP-tagged PIS precipitated on Ni-NTA column during the washing steps, probably due to delipidation effects. Therefore we do not have the data for PIS, unfortunately.

Fig. 2D. FSEC-TS profile of *tt*SI (filled) and hSI (open) using TGP-FSEC (black) and Trp-FSEC (red). Apparent T_m values (°C) are indicated in the figure panel.

Minor concerns

1. Page 1. “Current strategies for membrane protein (MP) structure studies require functional isolation in large quantities but their hydrophobicity brings great challenges in almost every step such as expression, purification, and crystallization”.

The authors should mention Cryo-EM techniques as a method for structure determination by citing a couple of articles, as Cryo-EM is becoming more common these days.

-- Done

2. If there is no patent or license issue, I highly recommend the authors to deposit their vectors (pETSG, pYTSG, pBTSG, Sb44, Sb66 and Sb68) to Addgene so that their method can be more widely-used. I believe it would highly benefit the community. After deposition, the authors should show the Addgene IDs in the revised manuscript.

-- **We thank the reviewer for the suggestion to make our work potentially more visible.**

A statement has been added to the revised manuscript under ‘Data availability’ in the Material & Methods:

Relevant plasmids and sequences have been deposited in Addgene (www.addgene.org) with the following IDs: pETSG, 159418; pYTSG, 159419; pBTSG, 159420; pSB_init_Sb44, 159421; pSB_init_Sb66, 159422; pSB_init_Sb68, 159423; pSB_init_Sb92, 159424.

The vectors for insect cell expression are in the process of depositing. We plan to include their Addgene IDs at a later stage.

Reviewer #3 (Remarks to the Author):

In this manuscript by Cai Yao and Li describes the use of thermostable GFP (TGP) as a fusion protein for membrane proteins (MP). Fusion of a fluorescent protein is useful in structural studies where scanning multiple variables such as species choice, construct design, extraction detergent, and buffer is required in order to boost protein expression levels and enable crystallization. Since stable proteins tend to crystallize better, it is advantageous to identify mutations that increase the thermostability of the MP. When testing the stability of each construct and each condition it is beneficial to have a thermostable fluorescent fusion protein that will report on the state of the MP rather than undergo unfolding itself. In this study the authors applied a new fluorescent protein, TGP, to the current system of fluorescence detection size exclusion chromatography thermostability assay. The authors tested TGP as a fusion protein in three expression systems, mammalian, yeast, and bacteria and showed that in addition to faithfully report on the T_m of the MP, the fusion also help increase expression levels as compared to the same MP fused to GFP. The authored also generated synthetic nanobodies (sybodies) against TGP to facilitate purification. The authors also solved a crystal structure of TGP in complex with one of these sybodies.

This study provides a method advance to allow selection of more thermostable membrane protein primarily for structural studies. There are some technical points to consider, but otherwise the study is well-executed and does not overstate its finding.

-- We thank the reviewer for the favorable summary and specific suggestions for improving our manuscript.

Major points:

1. The study was done in three expression systems, mammalian, yeast, and bacteria. While mammalian cell and bacteria are frequently used in structural biology, hardly any largescale protein production is done in *Saccharomyces cerevisiae*. In order for this study to be truly useful for the structural biology community it would be better to choose a more widely used expression system for membrane proteins such as sf9 insect cells. There are advantages to yeast, such as the ease of cloning, but even though generating baculovirus is more demanding it will better reflect how this method will be used in a lab.

-- Following the reviewer's comments to make the study more complete, we performed the comparison in the insect cell system. The results showed that replacing scGFP by TGP consistently improved expression level of hSI and PIS in insect cells. The following section has been added into the revised manuscript.

Replacing GFP with TGP improved MP expression - *insect cells*

When tested in insect cells, replacing scGFP with TGP increased the expression of hSI (Fig. 5A-5D) by 2 folds, based on the in-gel fluorescence (Fig. 5B) and Yield_{FSEC} analysis (Fig. 5C, Table 1). This trend was also observed for PIS, which showed a 1-fold increase when replacing scGFP with TGP (Fig. 5E-5H, Table 1). Again, the FSEC profile of the MPs were

the same regardless of the FP tags, although more fractional degradation was observed for the TGP-tagged MPs.

Fig. 5. Replacing GFP with TGP improved expression of two human MPs in insect cells. Assessment was based on fluorescence counts (A, E), in-gel fluorescence (B, F), and relative FSEC intensities (C, G). Normalized FSEC traces are shown in (D, H). Data are either from all (A, E), or a representative (B, C, F, G), of three independent experiments on different cell dishes.

2. The authors used Isothermal titration calorimetry (ITC) to measure the sybodies affinity to TGP. While ITC has its merits, the gold-standard technique for measuring affinity of antibodies and antibody-fragments is Surface Plasmon Resonance (SPR). In addition to K_d , K_{off} and K_{on} are also measured thus providing additional important kinetic information. This will help in selecting a superior sybody that shows both a long K_{off} and low K_d for TGP.

-- We have measured the binding kinetics of the sybodies for TGP binding using the biolayer interferometry (BLI) assay.

While the K_D values were consistent with the ITC values for some, that for Sb44 was very different (ITC, 109 nM; BLI, 4.4 nM). This may have been caused by the nature of the two different methods. We have replaced all the ITC results with the BLI results.

The following sentence describe the BLI results in the revised manuscript.

Biolayer interferometry (BLI) assays showed that the sybodies bound to TGP with affinities in the range of 3.3-10.4 nM (Fig. 7C-7F).

Minor points:

1. The authors show that fusion of TGP gives higher yield as compared to fusion to GFP. It would be useful to show yields of the unfused MP for comparison. Some fusion proteins actually interfere with proper folding of their partner and thus give lower yields.

-- We thank the reviewer for sharing his/her knowledge and experience in protein expression with tags. We have observed this too in the past (*Protein Expr Purif* 2019 164: 105463). In that study, the GFP-free hSI had much higher expression level than the hSI-GFP.

On the contrary, however, fusing a tag (MBP, GST, TGP) to membrane proteins sometimes increase yield. Thus, the effect of FP tags on protein expression can vary between different POIs. Because of the uncertain effect, and because determining expression level without FPs can be time consuming, we did not report careful comparison for TGP-free and TGP-tagged POIs in this paper. Instead, we have added the following paragraph to the Discussion.

The increase of POI expression by the *replacement* of TGP for GFP should not be interpreted such that the TGP tag always increases expression of POIs in comparison with their tag-free forms. Including a chaperon may be beneficial for heterologous expression, but this will inevitably increase energy expenditure of the host. In addition, depending on the POIs, tags may interfere with POI's folding and localization, and hence may influence negatively on production yield¹⁴. Therefore, the effect of FP tags on membrane protein expression should be tested on a case-by-case basis, when needed.

2. In the comparison of the melting profile of spPlsY using both enzymatic and fluorescence it is mentioned that the enzymatic activity dropped gradually upon heating (which is typical) while the fluorescence increased before dropping (lines 125-132 and Fig.2E). This was done only once and so there are no error bars. This experiment should be repeated a few times to generate error bars and asserting whether this is just a one-time occurrence or is this an actual phenomenon. This will provide the audience of this paper with some knowledge when they try to implement this method.

-- We have repeated this experiment for spPlsY and updated the results in Fig. 2E. The profile was reproducible. We have added the following sentence in the Figure legends.

In E, the FSEC data are from four independent experiments.

The increase of fluorescence before dropping is not always apparent but quite often, as we discusses in the following paragraph.

The FSEC-TS trace increases before dropping for some TGP fusion proteins (Fig. 2B, 2E, 2F),

but not for all (**Fig. 2A, 2C, 2D**), a phenomenon also seen in literature^{40,43,61}. Because the increase was not observed for the free TGP, and because it appeared at different temperatures for different MPs, we propose it is not intrinsic for TGP. Instead, it probably reflects the folding states of POI. It has been reported that GFP fluorescence can respond to subtle environmental differences caused by POI unfolding (before aggregation), and the resulted changes in intensity (decrease in those cases) can be used directly to monitor unfolding by differential scanning fluorimetry⁷⁷. In our assays, the rise-phase, appearing just before the drop-phase (aggregation), might also reflect local denaturation of POIs before aggregation.

3. The selection of sybodies generated three viable options. The authors chose Sb44 which, at $K_d=109$ nM has the worst affinity for TGP. If there were other considerations for this choice, such as poor yield or folding issues for the other sybodies, they should be explicitly stated.

-- All the sybodies could be expressed and purified to relatively high yields (Sb44, 48 mg L⁻¹; Sb66, 10 mg L⁻¹). We chose Sb44 because it was the first one been characterized and purified.

The yield information has been updated in the revised manuscript as the sentence below. We again thank the reviewer for catching this important information.

The sybodies could all be purified from *E. coli* with relatively high yields ranging from 10-48 mg L⁻¹.

4. In the paragraph between lines 407-412 the authors suggest testing whether the sybodies bind to the same epitope. The authors also suggest two very straight forward and simple experiments to test this hypothesis. I encourage the authors to pursue this, since as they state in the same paragraph, this can be a very useful piece of information.

-- We performed FSEC screening and identified Sb92-Sb66 as a non-competing pair.

The new results have been added to the revised manuscript.

Nanobodies recognizing different parts of GFPs have been reported in literature⁷⁸. Such nanobodies could be used to construct bi-paratopic fusion to enhance affinity^{70,78}. To seek such nanobodies for TGP, we performed FSEC analysis of the Sb66-TGP complex in the presence of other binders. This identified Sb92 as a non-competing partner for Sb66 for TGP-binding (**Fig. 7H**).

5. In the Methods section in lines 458-465 the contents of the lysis buffer are not stated.

-- Done. The buffer contains 150 mM NaCl and 50 mM Tris HCl pH 8.0.

6. For clarity, I think it is best to change the title in line 554 (Stability assay-TGP-or GFPbased FSEC-TS) to “membrane isolation and protein purification” or something along these lines for the paragraphs between lines 555-573. The original title should be applied to the paragraph between lines 575-586.

-- **Done with thanks.**

7. Line 594 change “was” to “were”

-- **We thank the reviewer for catching this and the following errors.**

8. Line 709 change “gravitation” to “gravity”

-- **Done.**

9. In lines 734-737 the authors describe preparation of affinity resin by coupling Sb44 to CNBr resin. To make this point clear to the reader the section should start with something like “Sb44 resin was prepared by ...”

-- **Done.**

10. In line 766 it is not clear that the 70 ul refers to the volume of the reservoir. The authors should change the phrasing to make this clear to the reader.

-- **The sentence was re-written as the following.**

Sitting drop crystallization was performed by first pipetting 70 μL of precipitant solution into each well as reservoir, followed by depositing 150 nL of the precipitant solution on top of 150 nL of protein with a Crystal Gryphon LCP robot (Art Robbins Instruments).

REVIEWERS' COMMENTS:

Reviewer #1 (Remarks to the Author):

I am satisfied with the authors response to my previous concerns and have no further comments.

Reviewer #2 (Remarks to the Author):

The authors properly addressed most of my concerns. Now I only have a few minor requests for textual changes regarding one of the updated parts in the revised manuscript.

Page 13-14. "Occasionally, membrane proteins, even for those with a high apparent T_m in the FP-fused form, precipitate upon FP removal, suggesting protective effect of these stable tags on membrane proteins. Such effects, if not affecting membrane protein fold and activity, may be advantageous for cryo-EM studies because density from flexible tags could be averaged out during reconstruction⁸³ so the data may be acquired using FP-tagged protein. For crystallization, it may require changing to FP-free constructs so that the POI did not experience compositional shock during later purification steps."

"Occasionally, membrane proteins, even for those with a high apparent T_m in the FP-fused form, precipitate upon FP removal, suggesting protective effect of these stable tags on membrane proteins."

In this discussion section, the authors should mention their results on hSI (decreased T_m without TGP) and PIS (precipitation after the removal of TGP) again. They should also cite other papers describing a similar potential pitfall, for instance doi: 10.1038/nprot.2008.44 and doi:10.1101/2020.09.28.316307.

"Such effects, if not affecting membrane protein fold and activity, may be advantageous for cryo-EM studies because density from flexible tags could be averaged out during reconstruction⁸³ so the data may be acquired using FP-tagged protein."

As far as I know and based on the paper cited by the authors(ref 83. With megabody), there is no successful report of structure determinations of GFP-fusion membrane proteins by cryo-EM. The authors should mention it. Otherwise, they can simply remove this sentence.

Reviewer #3 (Remarks to the Author):

All of my concerns were met.
I support publication of this manuscript.

Reviewer #1 (Remarks to the Author):

** I am satisfied with the authors response to my previous concerns and have no further comments.

We thank the reviewer for the previous comments to improve our manuscript.

Reviewer #2 (Remarks to the Author):

** The authors properly addressed most of my concerns. Now I only have a few minor requests for textual changes regarding one of the updated parts in the revised manuscript.

We thank the review for the careful reading and the specific comments below to improve clarity of our manuscript.

** Page 13-14. “Occasionally, membrane proteins, even for those with a high apparent T_m in the FP-fused form, precipitate upon FP removal, suggesting protective effect of these stable tags on membrane proteins. Such effects, if not affecting membrane protein fold and activity, may be advantageous for cryo-EM studies because density from flexible tags could be averaged out during reconstruction⁸³ so the data may be acquired using FP-tagged protein. For crystallization, it may require changing to FP-free constructs so that the POI did not experience compositional shock during later purification steps.”

“Occasionally, membrane proteins, even for those with a high apparent T_m in the FP-fused form, precipitate upon FP removal, suggesting protective effect of these stable tags on membrane proteins.”

In this discussion section, the authors should mention their results on hSI (decreased T_m without TGP) and PIS (precipitation after the removal of TGP) again.

The T_m of hSI did not decrease after removing TGP. Results in Fig. 2d showed that the TGP-fused and TGP-free form of hSI had very similar T_m (31.2 versus 30.5 °C).

The decrease the reviewer refers to is probably the T_m of hSI-TGP *after* purification (Fig. 2d, 31.2 °C) compared with that *before* purification (Fig. 2c, 49.4 °C). We speculate that the decrease was caused by delipidation during the purification.

We realize that the confusion might have been caused by lack of clarity in the writing and/or in the labels in Fig. 2c/d. We therefore re-wrote the paragraph as the following:

“Second, we compared the FSEC-TS profile of membrane proteins with and without TGP fusion using purified samples which are necessary for the tryptophan-based assay. For this we

tested *hSI* and its homolog from *Thermothlomyces thermophilus* (*ttSI*)⁶⁴. As shown in **Fig. 2d**, the TGP-based pseudo-melting curve for the fusion protein was nearly superimposable with the tryptophan-based curve for the TGP-free protein in both of the cases. Therefore, the TGP fusion did not change the apparent T_m of *ttSI* or *hSI*.”

Fig. 2. FSEC-TS of membrane proteins using different methods. (c) Pseudomelting curve of non-purified *hSI* tagged with TGP (black) or with scGFP (red). (d) Pseudomelting curve of purified *ttSI*-TGP (black closed circle), TGP-free *ttSI* (red close square), *hSI*-TGP (black open circle), and TGP-free *hSI* (red open square).

We also changed the labels in Fig. 2d by adding “-TGP” to the fusion constructs to better reflect the different protein forms.

Likewise, it was the PIS-TGP protein that precipitated during the purification.

We added a sentence in the Fig. 2 legends (“The T_m difference of *hSI*-TGP between c and d is likely due to delipidation effect (see Discussion).”) to direct readers for the Discussion of the delipidation effect:

“We note a 18-°C decrease of the apparent T_m for *hSI*-TGP after purification (**Fig. 2d**, 31.2 °C) compared with that obtained using the membrane solubilization fraction (**Fig. 2c**, 49.4 °C). This is probably caused by the likely delipidation events during chromatography steps. The loss of stability after purification was also observed for PIS, in which case the PIS-TGP fusion protein precipitated during our attempts in isolating pure TGP-fused and TGP-free PIS for stability comparison as was done for *ttSI* and *hSI* (**Fig. 2d**).”

** They should also cite other papers describing a similar potential pitfall, for instance doi: 10.1038/nprot.2008.44 and doi:10.1101/2020.09.28.316307.

Done.

** “Such effects, if not affecting membrane protein fold and activity, may be advantageous for cryo-EM studies because density from flexible tags could be averaged out during

reconstruction⁸³ so the data may be acquired using FP-tagged protein.”

As far as I know and based on the paper cited by the authors (ref 83. With megabody), there is no successful report of structure determinations of GFP-fusion membrane proteins by cryo-EM. The authors should mention it. Otherwise, they can simply remove this sentence.

We have removed this sentence in the revised manuscript.

Reviewer #3 (Remarks to the Author):

** All of my concerns were met.

I support publication of this manuscript.

We thank the review for the support of our work.